# Computational multiphysics modeling of radioactive aerosol deposition in diverse human respiratory tract geometries
Ignacio R. Bartol [1], Martin S. Graffigna Palomba[1], Mauricio E. Tano [2,3] ✉ & Shaheen A. Dewji [1,3] ✉

The evaluation of aerosol exposure relies on generic mathematical models that assume uniform particle deposition profiles over the human respiratory tract and do not account for subject-specific characteristics. Here we introduce a hybrid-automated computational workflow that generates personalized particle deposition profiles in 3D reconstructed human airways from computed tomography scans using Computational Fluid and Particle Dynamics simulations. This is the first large-scale study to consider realistic airways variability, where 380 lower and 40 upper human respiratory tract 3D geometries are reconstructed and parameterized. The data is clustered into nine groups using random forest regression. Computational fluid and particle dynamics simulations are conducted on these representative geometries using a realistic heavy-breathing respiratory cycle and radioactive iodine-131 as a source term. Monte Carlo radiation transport simulations are performed to obtain detailed energy deposition maps. Our findings emphasize the importance of personalized studies, as minor respiratory tract variations notably influence deposition patterns rather than global parameters of the lower airways, observing more than 30% variance in the mass deposition fraction.

The prevailing framework for radiation dosimetry from the inhalation of airborne radioactive particles relies heavily on mathematical models from the International Commission on Radiological Protection (ICRP)[1–3]. In these models, the human respiratory tract (HRT) is segmented into a series of representative compartments, assuming that particle deposition is uniform within those compartments. While ICRP models have proven to be appropriate for prospective radiation protection purposes based on scaling factors[4], inclusive of particle size distributions for internal intakes via inhalation, these models lack physiologically-specific non-phenotypic information related to the HRT, such as trachea diameter, lung volume, bronchi length, and other parameters.

While rare, historical incidents have involved the accidental release of particulate radioactive material into the environment, e.g., nuclear weapon tests[5], nuclear power plant (NPP) accidents[6,7], and NPP decommissioning[8].

Additionally, environmental radon progeny pose a potential health risk from adsorption onto airborne particulate matter, facilitating pulmonary deposition upon inhalation and consequent radiological damage to lung tissue[9]. Lastly, in the domain of nuclear medicine, aerosolized radionuclide, such as $^{131}I$, or institutions that generate $^{224}Ra/^{212}Pb$, pose a distinct challenge, necessitating stringent control measures to mitigate the risk of internal contamination[10].

The ICRP develops guidance for prospective radiation protection focused on harnessing reference anatomical models (50th percentile) and sex-averaged metabolic models in the development of dose quantities[11]. Furthermore, the ICRP HRT assumes a single homogeneous lung airway, parenchyma, and vascular composition without definition of lower generations and source distribution in the lung in the computation of absorbed and equivalent doses, and radionuclide S-values[3], defined as the average absorbed dose to a target region per nuclear decay in the source volume. In reality, a potentially exposed population comprises of both sexes and a plurality of ages. Subject-specific or sub-population models are, therefore, required to appropriately evaluate the impact (i.e., dose) of inhaled aerosolized radioactive particles, spanning occupational workers, members of the public, and first responders.

In this study, a hybrid automated workflow was developed to advance the field of internal dosimetry to better represent population phenotypes and source particle distributions deposited in the lungs from an inhaled radionuclide pathway. This workflow serves as a physiologically-specific complement to existing ICRP compartment-based models, where the approach developed herein incorporates a subject-specific particle deposition data in the HRT through the definition of a realistic airway geometry. Utilizing subject-specific Computed Tomography (CT) scans to define a

[1]Georgia Institute of Technology, Atlanta, GA, USA. [2]Idaho National Laboratory, Idaho Falls, ID, USA. [3]These authors jointly supervised this work: Mauricio E. Tano, Shaheen A. Dewji. ✉e-mail: mauricio.tanoretamales@inl.gov; shaheen.dewji@gatech.edu

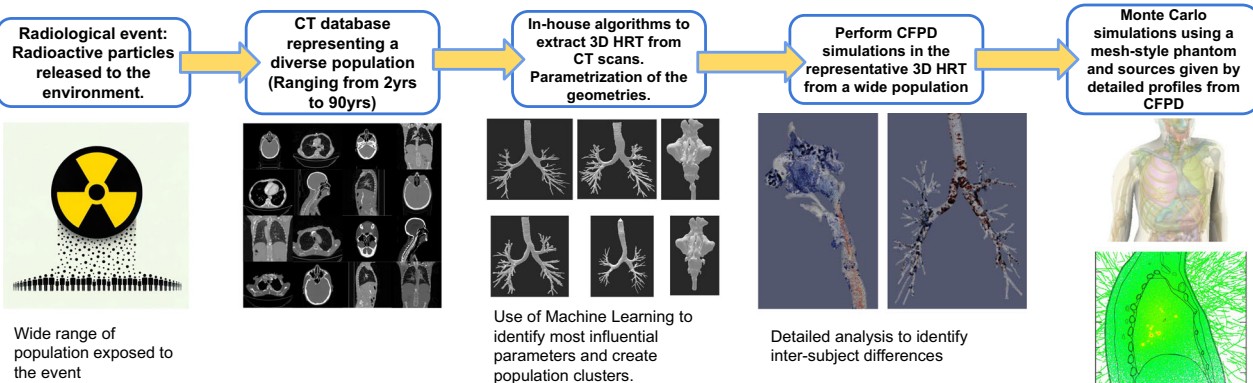

**Fig. 1 | Schematic representation of the workflow and research presented in this study.** This figure highlights the automated pipeline for reconstructing and parametrization of HRT geometries from Computed Tomography (CT) scans for comprehensive airflow, particle deposition, and radiation dosimetry analysis. The use of Random Forest for feature identification, kernelized k-means for clustering Human Respiratory Tract (HRT) geometries, and algorithmic integration of Computational Fluid and Particle Dynamics (CFPD) simulations with the PHITS (Particle and Heavy Ion Transport code System) Monte Carlo radiation transport tool for subject-specific internal dosimetry, illustrated for a large cohort of de-identified subject data, representative of the breadth of an exposed population. The overarching goals of this research are to: (1) establish a streamlined workflow that rapidly converts CT scans into 3D models of the human respiratory tract for CFPD simulations; (2) enhance the fidelity of CFPD simulations by incorporating representative particle source terms and boundary conditions; (2) perform subject-specific analysis in the particle deposition profiles, leading to the generation of accurate radioactive dose maps using mesh-style phantoms within Monte Carlo simulations; (4)and apply the developed CFPD workflow and simulation tools on a broad and representative population to identify inter-subject differences in radioactive particle deposition.

high-fidelity 3D anatomical geometry using computer vision algorithms, Computational Fluid and Particle Dynamics (CFPD) were employed to generate particle deposition profiles (PDPs), improving upon ICRP assumptions of a homogeneous anatomical definition and empirically-derived deposition equations implemented as a homogeneously distributed volume source in the lungs. These profiles were then integrated with a Monte Carlo radiation transport code to characterize the distribution of absorbed dose within the lungs. Figure 1 depicts the workflow of the hybrid automated framework conducted in this study.

Computational Fluid Dynamics (CFD) is the foundational approach for modeling airflow in the HRT proposed in this study. By exploiting the Navier-Stokes equations, different models investigate different effects on human airways. Specifically, CFD has the advantage of visualizing local phenomena that cannot be seen otherwise by conducting in-vitro or in-vivo experiments. A plurality of studies employing CFD have analyzed intricacies of the nasal airflow under different breathing conditions using realistic nasal models[12], assessing the airflow in subjects with rhinosinusitis[13], and investigated how the laryngeal jet generated in the trachea affects the airflow downstream in the intrathoracic human airways[14]. When particles are introduced into the system, the Navier-Stokes equation for the flow field needs to be coupled with the equations of motion for the particles. To effectively model aerosol transport in the human airways, consideration of the aerodynamic size of particulate matter is required to determine the interaction mechanism. Particles ranging from 10 nm to 100 μm may be represented as solid or droplet phase geometries and may change size due to evaporation or condensation. The high humidity in the respiratory tract can cause hygroscopic particles to grow. Particles undergo various physical interactions with the fluid, leading to different deposition mechanisms along airway surfaces, including impaction, sedimentation, and diffusion. The impaction mechanism affects particles with sufficient momentum, causing deposition at airway branches due to abrupt directional changes in airflow. Sedimentation occurs when particles with enough mass deposit due to gravity after traveling in the airways. The Brownian motion mechanism causes deposition by random diffusion, particularly for nano-sized particles.

In transitioning from CFD to CFPD, coupling the Navier-Stokes equations with the equations of motion for particles is critical for simulating fluid-particle interactions, accounting for the various mechanisms of particle deposition influenced by airflow and different particle sizes. When coupling the flow field with particles, there are two primary approaches to consider. The first approach is using a Lagrangian frame of reference, which treats particles as discrete points moving based on interactions within the flow field and other particles, by solving the equations of motion for each particle. The second approach is using an Eulerian frame of reference, which assumes particles are a continuous medium with representative properties, by solving a convection-diffusion equation. In both cases, the flow field is resolved on an Eulerian description[15]. In the following sections, the terms Euler-Lagrange and Euler-Euler refer to the method used to solve the motion of the particles, where the first word in the term references how the flow field is solved, and the second word refers to the method used to solve the motion of particles.

On the other hand, there are different levels of complexity on how to model the particle and fluid interactions. The one-way coupled Euler-Lagrange method assumes particles do not affect the carrier flow, meaning particles move without providing any feedback to the airflow. The two-way coupled Euler-Lagrange method considers the impact of particles on the fluid motion, requiring a detailed coupling of particle forces into the airflow momentum equations. For more complex interactions, such as particle-particle collisions, the four-way coupled Euler-Lagrange method adds terms to account for these interactions[16]. An alternative Euler-Euler approach, suitable for spherical and quasi-spherical particles smaller than 100 nm, uses mass transfer equations to model particle behavior, assuming isotropic diffusion and turbulent dispersion.

The deposition of aerosolized particles in the HRT continues to remain a field of active investigation spanning multiple application spaces, such as turbulent effects[14,17], integration with radiation dosimetry codes[18], drug delivery mechanisms[15,19], under different respiratory conditions[20], and the comparison between monodispersed and polydispersed particle size distributions[21], among others. State-of-the-art approaches in simulating aerosol deposition in the HRT have employed high-fidelity geometry models[20,22,23]. Typically, each CFPD or CFD study begins by constructing a realistic or approximated 3D model of the HRT, either based on CT scans[24-27] or morphometric data[28,29]. Following the selection of the geometry, selecting a suitable turbulence modeling technique is critical to accurately resolve the complex flow phenomena—ranging from the laryngeal jet to the relaminarization in the lower bronchi. In this regard, Reynolds-Averaged Navier-Stokes (RANS) has proven to be as effective as Large-Eddy Simulation (LES) models[21,30,31].

RANS models average the effect of the turbulence, while LES models resolve large-scale turbulent structures directly. LES methodology uses filtered Navier-Stokes equations and models only the smaller scales of turbulence, hence the less energetic ones. Koullapis P. G. et al.[32], demonstrated that certain RANS models are as effective as LES simulations for predicting turbulence effects and particle deposition patterns, but RANS slightly overpredicting particle deposition in some cases. In the mentioned study, computational deposition measurements employed particle diameters ranging from $d_p = 0.5\,\mu m$ to $10\,\mu m$, particle density $\rho_P = 914\,kg\,m^{-3}$, and a constant airflow of $Q = 60\,L\,min^{-1}$, $Q = 30\,L\,min^{-1}$, and $Q = 15\,L\,min^{-1}$. The main constraint for LES models is that the computational cost is outstandingly higher than RANS models. LES requires finer meshes and smaller temporal resolution, increasing computational time by at least an order or magnitude. This increase in computational time makes LES turbulence models less feasible for routine clinical applications or large-scale studies.

While remarkable advancements have been made in integrating CFPD with more realistic HRT models to predict higher fidelity PDPs, the current state-of-the-art lacks the ability to conduct personalized studies in a timely manner and using representative boundary conditions of a realistic respiratory cycle. Therefore, a need exists to perform subject-specific studies in an automatized manner, optimizing computing time while increasing the precision of the particle deposition in the HRT. An automated methodology also provides accurate internal radiation dosimetry for the exposed public that lies outside the average population usually represented in dosimetric models[1]. Therefore, this study first introduces a hybrid-automatized workflow to perform subject-specific CFPD in the HRT and, second, provides insight into why subject-specific dosimetry tools are needed by comparing particle deposition across different HRT geometries.

To achieve the subject-specific dosimetry goals, integrating a synergistic CFPD-Monte Carlo radiation transport code is essential for forecasting individualized dose deposition patterns within pulmonary structures and adjacent organs. While Talaat K. et al.[18] developed a CFPD-MCNP tool coupling CFPD of internalized particles and radiation transport, this effort was limited to idealized reference geometries from ICRP, limited representation of the respiratory cycle, one-way coupling between the flow field and for the particles, and restricted to a low flow-rate. As a result, efforts have been made in the present study to incorporate realistic geometries and a representative respiratory cycle.

To achieve subject-specific level studies, a recently developed hybrid automated framework becomes necessary to harness accurate anatomical geometries of HRT models from CT scans with CFPD integrated with Monte Carlo radiation transport to represent population-specific physiological models for the estimation of deposition and subsequent estimation of inhaled radiation dose. This proposed hybrid-automated workflow uniquely harnesses multiphysics tools to model the mechanism of radiation-related response from the inhalation pathway to elucidate the etiology of radiation-induced health outcomes.

Furthermore, prior efforts employing CFPD have often emphasized the impact of uniform versus polydisperse aerosol particle sizes[21,22,33] or explored alterations in HRT geometry related to disorders like sleep apnea[31,34] and wall roughness[35]. The present study further expands the mechanism by which airflow and particle deposition vary with key HRT parameters, such as trachea length, the angle between the trachea and the main bronchi, and trachea diameter.

In the study explored herein focused on internal dosimetry calculations, particle distribution profiles from CFPD simulations are coupled with the Particle and Heavy Ion Transport code System (PHITS) Monte Carlo radiation transport code[36]. A 3D phantom was employed from ICRP Publication 145[11] mesh-type anthropomorphic phantoms of a reference adult male and female, improving upon the prior generation of anthropomorphic phantom modeling utilizing voxel-based phantom[18,37]. The latest generation of mesh-type phantoms offers multiple advantages, permitting the coupling and actuation of the hybrid workflow model. Prior generation voxel-based phantoms required substantial additional spatial resolution to accurately represent the complexities of different anatomical compartments, notably lacking the oral cavity, nasal cavity, trachea, and main bronchi. Such resolution deficits are a primary source of uncertainty in calculating the dose to different organs and distribution predictions, as corroborated by several comparative studies between the stylized and voxel phantoms[38–40].

Overall, the study presented here comprehensively analyzes HRT geometrical features, airflow, particle deposition, and radiation dosimetry across a large cohort of de-identified subjects who have undergone chest or head/neck CT scans from publicly available datasets. An automated pipeline was introduced to extract and parametrize 3D geometries of the lower or upper respiratory tract from CT scans. This work further incorporated the use of a Random Forest regressor algorithm, to identify the most important features of the HRT in the subject cohort. A kernelized k-means algorithm was also used to cluster the HRT geometries corresponding to different populations based on the relevant parameters found and simplify the dataset for the CFPD simulations from hundreds of HRT domains to only 18 representative geometries.

This first-of-a-kind pipeline also includes semi-automated post-processing to prepare the geometry for CFPD simulations and can generate the input files for either open-source software OpenFOAM[41] or licensed software StarCCM+ from Siemens[42]. An algorithmic integration using Python couples CFPD deposition profiles with the PHITS Monte Carlo tool was also developed. This work, therefore, aims to offer robust airflow analyses on parameterized HRTs and elucidate particle deposition behavior, harnessing a simplified tool-set for conducting CFPD simulations on realistic human airway geometries using representative inlet conditions and particle size distributions. Concurrently, a practical approach to subject-specific internal dosimetry is outlined, beginning with CT scans and incorporating realistic source terms from a radiological event. A graphical abstract is presented in Fig. 1. The main result of this study demonstrated the propagated impact of minor deformations in HRT geometry definition, substantially influencing the predicted particle deposition in realistic HRTs rather than the global parameters of the HRT, such as trachea diameter and main bronchi bifurcation angle. Differences as high as 30% in the mass deposition fraction were attributable to these geometrical variations across HRTs. Moreover, the difference is more pronounced between the left and right lung deposition fractions, where a difference of more than 60% between lobar regions was observed.

## Results

The findings of this study have been organized in the following sub-sections. The first sub-section discusses the outcomes of the automated reconstruction algorithm and clustering. Through the implementation of Random Forest regression, it was determined that trachea diameter and main bronchi branching angle constituted the most important features within the HRT geometries. A k-means algorithm, using the features found by the Random Forest, was employed to reduce the comprehensive database of HRT geometries for male and female adults to a set of 18 clusters, thereby pragmatically integrating CFPD analyses while preserving population diversity.

The second sub-section details evaluation criteria of the mesh independence study to evaluate the convergence of the simulations. Instead of relying on the number of elements in the mesh, the definition of a minimum base element size was determined to provide a more effective and universally applicable criterion. A base element size >1.5 mm was sufficient for achieving convergence in CFPD simulations that utilized a RANS approach for turbulence modeling.

In the third and fourth sections, the significance of subject-specific analyses and inter-subject variability were investigated by examining airflow structures and PDPs across a diverse array of HRT subjects, selected based on the k-means clustering. Minor variations in the HRT were observed to result in substantial imbalances in particle deposition and distinct airflow behavior in the tracheal downstream regions.

The final section discusses the importance of accounting for heterogeneity in absorbed radiation dose maps when analyzing realistic

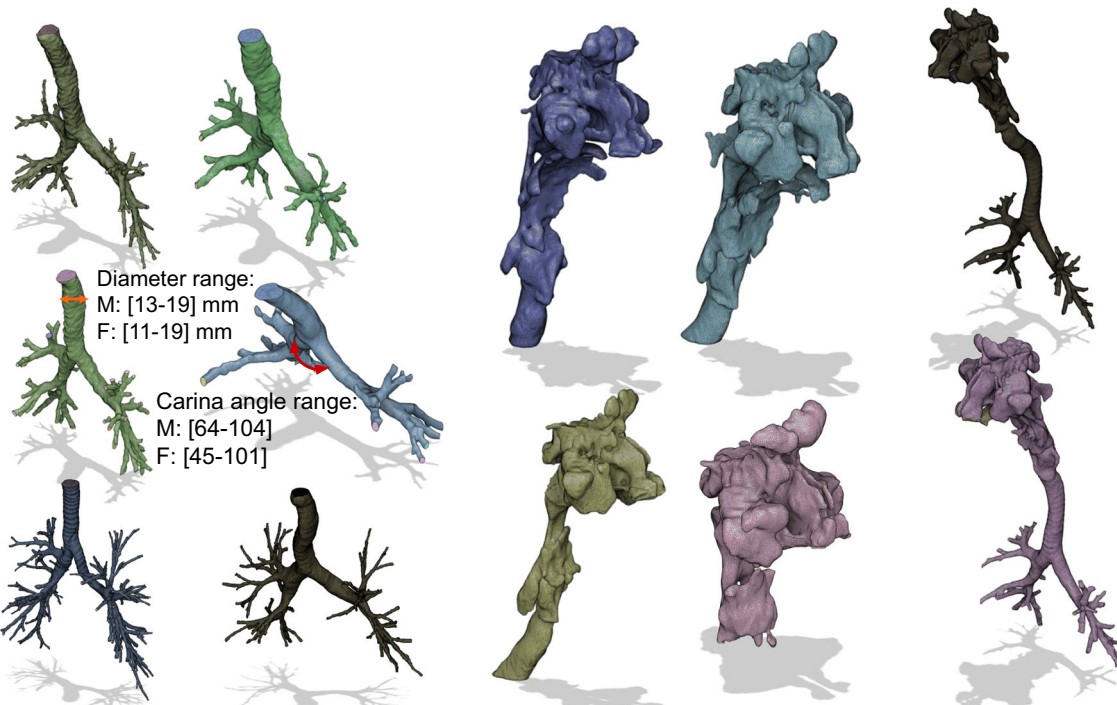

**Fig. 2 | Reconstructed 3D geometries of the Human Respiratory Tract.** A subset of segmented geometries derived from Computed Tomography (CT) scans is shown. A parametrization algorithm was applied to the lower part of the HRT, and the range of values for tracheal diameter and main bronchial angle is labeled for both female (F) and male (M) subjects.

radioactive PDPs. The CFPD simulations were coupled with the PHITS stochastic radiation transport code to achieve a harmonized and integrated multiphysics approach. By using a CFPD and Monte Carlo radiation transport coupling, detailed and precise maps of the absorbed radiation dose were obtained, improving upon the current practice of modeling source radiation in the HRT as a homogeneous volumetric distribution, while disparately estimating particle deposition using empirical equations. This multi-scale multiphysics integrated approach, folded with population phenotype considerations of the HRT, demonstrates a high-fidelity approach employing advanced computational tools to model the mechanism of radioactive particle transport and deposition, which will further inform the etiology of radiation-induced health outcomes from internalized radionuclides.

**Reconstruction Algorithm And Clustering**

The reconstruction algorithm, as described in the Methods section, was applied to CT scans, filtering out scans with low resolution or fewer than 100 slices. This resulted in 542 reconstructed 3D geometries of the lower HRT from subjects aged 2 to 90 years, with 54% males and 46% females, including 88 pediatric subjects. Of these, 412 geometries met the algorithm's criteria and were used in this study.

The algorithm achieved a 76% success rate, discarding anomalies like non-physical bronchi or unrealistic dimensions, a challenge also noted in previous studies[26,43–45]. For the upper HRT, 111 head-neck CT scans were segmented, with 40 geometries corrected for nostril leaks and nasal septum holes. To demonstrate proof-of-concept, four complete HRTs were connected using Blender[46]; the criteria for connecting the upper and lower HRT is further expounded in Supplementary Methods under the Geometry Pre-Processing section. A visual reference for the different reconstructed 3D geometries is visualized in Fig. 2.

In parallel with the process of geometry reconstruction, a specialized parametrization algorithm was executed to quantify critical anatomical features of the lower HRT. These features include volume-averaged tracheal diameter, main bronchi branching angles, overall segmented volume, and cumulative bronchial length. This step aimed to construct a comprehensive morphological spectrum of the HRT across a diverse population.

A Random Forest regressor was employed to discern the most pertinent features that can predict a subject's age and weight—information solely present in anonymized CT data sets; the relative importance of the parametrized features is illustrated in Fig. 3a. According to Fig. 3a, the tracheal diameter emerged as the most important feature, consistently ranking highest across multiple decision trees, closely followed by the angle of the primary bronchi. Subsequently, age and weight predictions, derived from the Random Forest model based on these anatomical parameters, are illustrated in Fig. 3b, highlighting the predictive capabilities of our method.

After performing feature importance selection, the dataset was visualized in its two most significant dimensions and subjected to clustering using a kernelized k-means algorithm. Specifically, a radial basis function was employed for the kernel, and standard normalization techniques were applied. Clusters were labeled based on their proximity to the mean values, facilitating the categorizing of the heterogeneous HRT landscape into a set of representative geometries. This discretization was conducted based on the normalized main bronchi angle bifurcation or carina angle ($b_a$) and normalized trachea diameter ($t_d$). The rationale for selecting nine clusters stems from stratifying the data into small, medium, and large values for each dimension. Figure 3c illustrates the clustering results for male subjects, categorizing them into nine distinct groups as detailed below. The specific characteristics of each group can be found in Table 1.

1. big $t_d$ and small $b_a$ (bs): This cluster represents subjects with a large tracheal diameter and small bronchial angle.
2. big $t_d$ and mean $b_a$ (bm): This cluster includes subjects with a large tracheal diameter and average bronchial angle.
3. big $t_d$ and big $b_a$ (bb): Subjects in this cluster have both a large tracheal diameter and large bronchial angle.
4. mean $t_d$ and big $b_a$ (mb): This cluster comprises subjects with an average tracheal diameter and a large bronchial angle.
5. mean $t_d$ and mean $b_a$ (mm): Subjects in this cluster have average values for both tracheal diameter and bronchial angle.

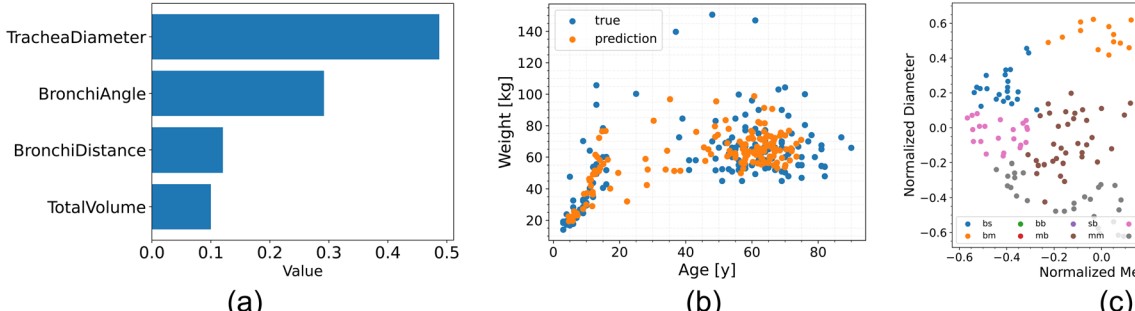

(a) (b) (c)

**Fig. 3 | Human respiratory tract geometry parametrization analysis and clustering. a** Relative importance of the phenotypical traits: trachea diameter, bronchi branching angle, total lung volume, and bronchi length, for predicting subject's age and weight. **b** Random Forest prediction on data, with root mean squared error less than 20%. **c** Normalized and kernelized trachea diameter versus main bronchi branching angle for male subjects, showing nine clusters selected using a kernelized k-means algorithm. Each label in this graph (bb, bm, bs, mb, mm, ms, sb, sm, ss) represent a cluster, which are defined based on tracheal diameter ($t_d$) and carina angle ($b_a$). The bs cluster represents subjects with a large $t_d$ and small $b_a$, the bm cluster is for large $t_d$ and average $b_a$. The bb cluster comprises subjects with both large $t_d$ and $b_a$. In the mb cluster, subjects have a mean $t_d$ and a large $b_a$, and the mm cluster represents mean values for both $t_d$ and $b_a$. Subjects in the ms cluster have an average $t_d$ and small $b_a$, while those in the sb cluster have a small $t_d$ and large $b_a$. The sm cluster includes small $t_d$ and average $b_a$, and finally, the ss cluster represents subjects with both small $t_d$ and $b_a$.

6. mean $t_d$ and small $b_a$ (ms): This cluster represents subjects with an average tracheal diameter and a small bronchial angle.
7. small $t_d$ and big $b_a$ (sb): Subjects in this cluster have a small tracheal diameter and a large bronchial angle.
8. small $t_d$ and mean $b_a$ (sm): This cluster includes subjects with a small tracheal diameter and an average bronchial angle.
9. small $t_d$ and small $b_a$ (ss): This cluster represents subjects with both a small tracheal diameter and small bronchial angle.

From this clusterization, one representative geometry from each cluster was selected to perform the CFPD simulations and get each PDP.

## Mesh Independence Study

In conducting a study on the independence of CFD meshing to validate simulation accuracy, a comprehensive verification and validation effort aligned with the National Program for Applications-Oriented Research in CFD (NPARC) Alliance Verification and Validation guidelines[47] was implemented. Seven meshes of varying fineness were created using a representative lower HRT geometry with seven bronchi generations. Simulations were conducted under a constant $Q = 90$ L min$^{-1}$ steady airflow. The remaining parameters for the airflow remained unchanged from Table 3, using the appropriate solvers.

Meshes ranged from a coarse 1.84 mm element base size to a fine 0.087 mm, with five additional meshes providing intermediate gradations, ensuring a refinement ratio between 1.25 and 2 as per NPARC standards. A high-fidelity simulation with a 0.054 mm element base size was performed to set a

benchmark for convergence. All the meshes were tested to pass mesh quality checks successfully, as given in Supplementary Fig. 1 in the Supplementary Notes 1, in the Verification and Validation section. In OpenFOAM, the element base size (i.e., mesh size) was defined by creating a background hexahedral mesh using the blockMesh tool. This tool sets the initial size of the hexahedral cells before any refinement or snapping processes. Similarly, the element base size in StarCCM+ was set as a global parameter that determines the initial tetrahedral cell size before any refinement to a polyhedral mesh or local sizing controls was applied. Convergence metrics — i.e., Order of Grid Convergence (calculated via a least-squares fit), Grid Convergence Index (GCI), and Asymptotic range of convergence — were used to assess the simulations. The computed convergence orders for pressure and turbulent kinetic energy can be seen in Supplementary Fig. 2 of the Supplementary Notes 1. Convergence orders were demonstrated to be marginally lower than the theoretical counterpart value of $p = 2$, attributable due to boundary conditions and mesh-specific errors.

GCI calculations for discretization error assessments were performed on three meshes, medium-to-fine, and coarse-to-medium grid analyses. Results yielded an error of less than 0.46% for pressure and less than 3.22% for turbulent kinetic energy, implying that the simulations had effectively reached grid independence within the asymptotic range. Therefore, it was determined that if the mesh has an element base size element of less than 1.5 mm, the simulation converged with less than 5% error from using a very fine mesh. Details regarding the GCI and mesh-specific details are given in Supplementary Table 1 and Supplementary Table 2, respectively, of the Supplemental Information. Details of the GCI study, including mesh configurations, mesh quality number of elements, and cell faces, can be located in the Supplementary Notes 1, Verification and Validation section.

## Airflow structures

The second phase of analysis in this study focused on the flow structures and secondary flows in both the lower and upper regions of the respiratory tract, sampled across a diverse male demographic. Figure 4 presents the main flow velocity profiles at various cross-sections for lower HRTs. These profiles are representative of subjects with large tracheal diameter and large branching angle in the primary bronchi, as delineated in Fig. 3c. Additionally, Fig. 4 illustrates the generation of secondary flows, expressed as tangential velocity vectors within the trachea.

The impact on particle deposition was explored through analysis of flow structures since the tangential velocity acts as a force that directs particles toward the wall of the respiratory tract. As emphasized in Fig. 4, the formation of large eddies occurs within both the trachea and primary bronchi. These large-scale turbulent structures are adequately captured by the $k - \omega$ Shear-Stress Transport (SST) Langtry-Menter (LM) RANS

## Table 1 | Range of carina angle and trachea diameter for each group in the k-mean clustering

| Group | Carina angle [degrees] | Mean trachea diameter [mm] |
|---|---|---|
| bs | 76.81 ± 2.42 (81.2-71.4) | 17 ± 0.5 (17.7-16.2) |
| bm | 80.9 ± 3.57 (87-76.4) | 17.3 ± 0.6 (18.2-16.1) |
| bb | 97.83 ± 3.53 (104-91.6) | 18.2 ± 0.3 (18.6-17.8) |
| mb | 95.18 ± 3.15 (101-90.6) | 17 ± 0.4 (17.6-16) |
| mm | 79.1 ± 3.5 (84.5-73.6) | 16.2 ± 0.4 (16.9-15.9) |
| ms | 69.23 ± 3 (73.2-64) | 14.9 ± 0.4 (15.9-14.3) |
| sb | 88.57 ± 3.6 (93.2-81.7) | 15 ± 0.5 (15.8-14) |
| sm | 82.65 ± 3.36 (89.7-78.4) | 14.3 ± 0.5 (15.5-13.7) |
| ss | 67.63 ± 3.2 (73.1-61.4) | 13.46 ± 0.1 (13.7-13.4) |

The mean value and the standard deviation are displayed in the table, the values in parenthesis are the minimum and maximum value within the cluster.

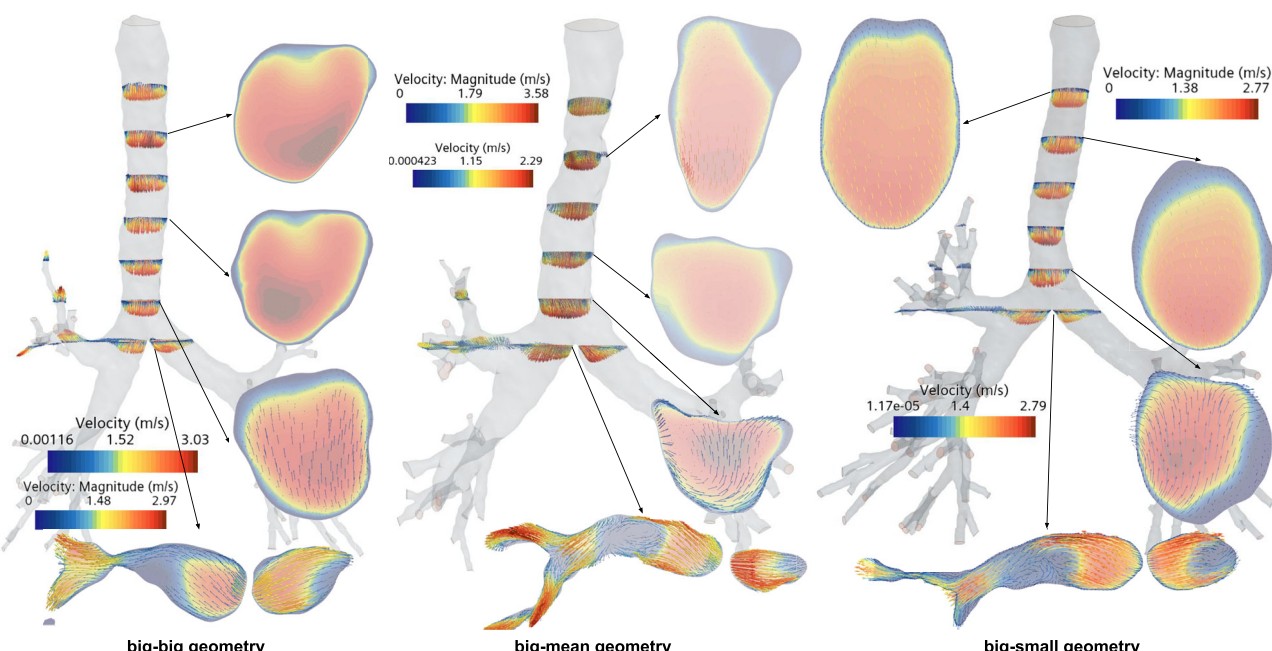

**Fig. 4 | Flow velocity profiles and secondary velocity flow in the different regions of the trachea and the main bronchi for a population subset from the clustering process.** This snapshot corresponds to the inhalation peak time step (i.e. 0.5 s in physical time). Specifically, the geometries displayed corresponds to the cluster of subjects with large trachea diameter and carina angle (big-big geometry, on the left); large trachea diameter and mean carina angle (big-mean geometry, on the middle); and large trachea diameter and small carina angle (big-small geometry, on the right).

turbulence model. However, the model lacks the granularity to resolve smaller eddies or the flow near the boundary layer, highlighting the importance of applying appropriate wall functions.

Additionally, minor geometric variations outstandingly influenced velocity fields even with identical inlet conditions in the eight HRT geometries tested. Varying airflow patterns impacted the calculated PDPs, thereby affecting particle distribution in the HRT. For instance, certain geometries feature stagnation points where flow circulation was absent, causing small particles to deposit via diffusion rather than impaction, contrary to common expectations in that region of the HRT. A notable asymmetry was also observed between the left and right branches of the main bronchi, which was consistent across various geometries. This asymmetry often gave rise to a large eddy in the left branch of the main bronchi.

Further images of the airflow structures in the rest of the geometries simulated and the full HRT (comprehending upper and lower parts) can be found in Supplementary Notes 1, Airflow structures and particle deposition, Supplementary Figs. 3 and 4.

**Particle deposition**
In the next analysis phase, variability in particle deposition patterns across representative geometries drawn from a diverse population sample was examined. Both the particle deposition fraction (nDF, Eq. (1)) and mass deposition fraction (mDF, Eq. (2)) were initially computed, followed by an assessment of the differences between the right and left lung lobes, defined in Eqs. (3) and (4). The particle deposition fraction computed in this work was corrected by ICRP Publication 130[3] deposition fractions in the alveolar region, accounting for particles escaping through the outlets in the inhalation cycle.

To proceed with the verification and validation of the method, the deposition efficiency was compared against data from prior studies[48–56], plotting the deposition efficiency as a function of the impaction parameter. The impaction parameter is defined as $d_p^2 Q$, where $d_p$ is the particle diameter and $Q$ is the airflow used. The rationale behind using the impaction parameter as a metric was that the particle deposition efficiency could be compared across different studies independently of the particle size and breathing conditions used. Ignoring potential numerical and experimental errors, the difference between studies were due to the different HRT geometries employed.

The studies listed in Fig. 5, from Zhou et al.[56] employed three different models and conducted both in-vitro (i.e. labeled as Exp) and in-silico (i.e. labeled as Num) experiments. The three models varied in complexity: USPIP, an idealized mouth-throat model (L-shaped tube); UofA, a simplified mouth-throat HRT; and LRRI, a realistic cast of the mouth-throat model. Arsalanloo et al.[49], performed CFPD in a realistic lower HRT but with a limited number of bronchi generations. Bowes S. M. et al.[50] conducted experimental work, measuring particle deposition in the mouth tract of human subjects, while Cheng et al.[51], performed in-vitro experiments in a realistic mouth+lower HRT cast, up to the 4th generation of bronchii. In the same vein, Kim, Y.H. et al.[52] conducted CFPD experiments in a realistic mouth-trachea plus lower HRT mode. Lastly, the results from Longest, et al[53], Matida et al.[54] and Zhang et al.[55], applied CFPD in studies using an idealized human mouth-throat model, while Liu et al.[48] used a combined mouth-throat and lower HRT model.

The airflow value used in this study for the verification, was selected as the mean airflow in the respiratory cycle. The results are displayed in Fig. 5, where good agreement is demonstrated against the verification dataset used. Values of $d_p^2 Q < 0.1$ were excluded from reported results due to lack of validation comparison data availability.

A comparison with Dong, J. et al.[57] was conducted to further validate the CFPD methodology. The study by Dong et al.[57] was an appropriate comparative validation, having employed a very similar 3D HRT model to one of the four full HRT geometries reconstructed in this work, in addition to a similar turbulence model and boundary conditions. The simulation parameters listed in Fig. 6, were adopted to mirror the conditions in the CFPD experiment from Dong, J. et al.[57], including the particle physics and coupling method. The results from the validation can be seen in Fig. 6. The boundary conditions were set to match Dong's et al.[57] work, where they used a constant inflow rate of 50 L min⁻¹ with a 1 s injection and 2 s breathing cycle. Simulation parameters were adapted to match those used in Dong et al.[57], including a particle density of 1000 kg m⁻³, one-way coupled particle-turbulence interactions, and the inclusion of drag and Brownian forces. The experimental setup and turbulence model (k-ω SST) were also replicated to ensure accurate comparison.

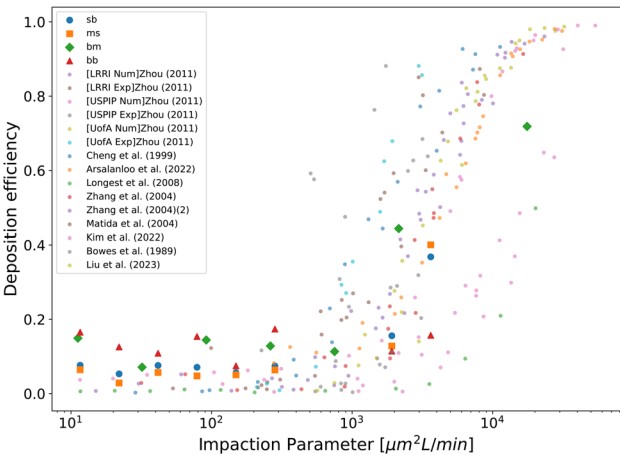

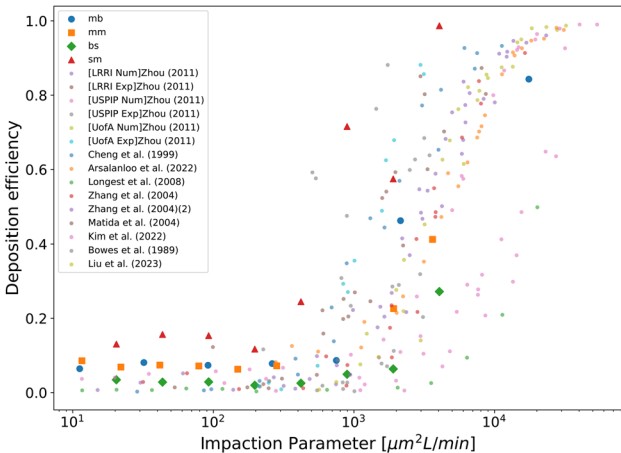

**Fig. 5 | Validation of the Computational Fluid and Particle Dynamics methodology by comparing deposition efficiency as a function of the impaction parameter ($d_p^2 Q$).** Left: cases 1–4 of 8 cases. Right: cases 5–8 of 8 cases. Each series (bb, bm, bs, mb, mm, ms, sb, sm, ss) represents a cluster, which are defined based on tracheal diameter ($t_d$) and carina angle ($b_a$). The bs cluster represents subjects with a large $t_d$ and small $b_a$, while the bm cluster includes those with a large $t_d$ and average $b_a$. The bb

cluster comprises subjects with both large $t_d$ and $b_a$. In the mb cluster, subjects have an average $t_d$ and a large $b_a$, and the mm cluster represents average values for both $t_d$ and $b_a$. Subjects in the ms cluster have an average $t_d$ and small $b_a$, while those in the sb cluster have a small $t_d$ and large $b_a$. The sm cluster includes small $t_d$ and average $b_a$, and finally, the ss cluster represents subjects with both small $t_d$ and $b_a$. Values of $d_p^2 Q < 0.1$ were excluded due to lack of validation comparison data availability.

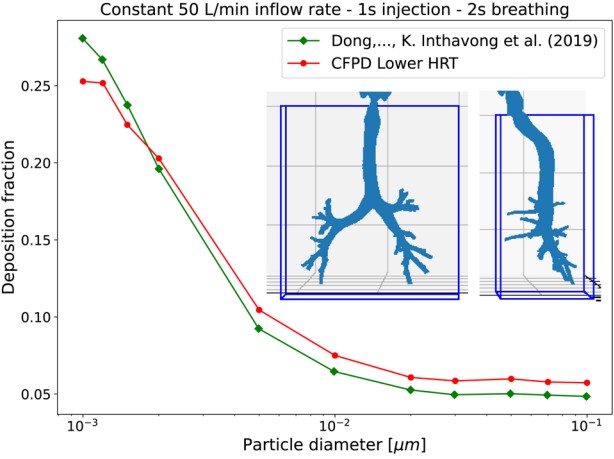

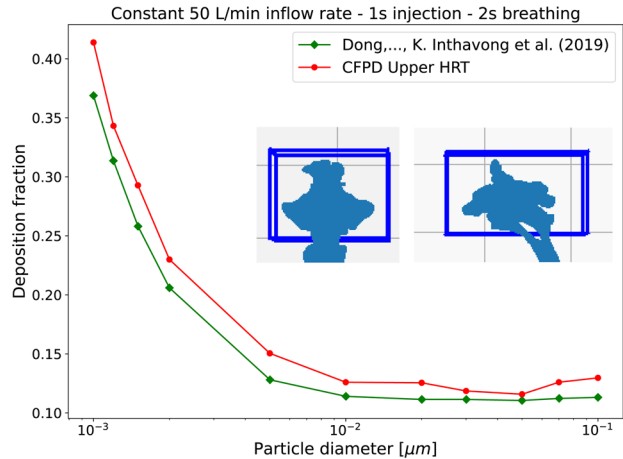

**Fig. 6 | Comparison of particle deposition fractions between one Human Respiratory Tract (HRT) from this study and the results of Dong et al.[57] using Computational Fluid and Particle dynamics (CFPD).** The left graph shows the deposition fractions for the lower HRT, while the right graph shows the deposition fractions for the upper HRT. The most relevant parameters used in the CFPD

simulation are defined as: constant inflow rate of 50 L min$^{-1}$ with a 1 s injection and 2 s breathing cycle, turbulence was modeled using Reynolds Averaged Navier-Stokes k-$\omega$ Shear-Stress Transport model, particle density of 1000 kg m$^{-3}$, one-way coupling between the airflow and particles, air temperature of 293 K, ~100,000 particles per particle size injected, and drag and Brownian forces were modeled.

Table 2 summarizes the tallied values of particle deposition from various geometries. The 3D model corresponding to the cluster of subjects with small tracheal diameters and small main bronchi angles was excluded from the analysis. This exclusion was due to the cluster's proximity to two other clusters, making it challenging to identify an evident representative HRT model within the presented dataset for that particular group. This exclusion will make eight HRT geometries where CFPD simulations will be run. To quantify the particle deposition, particle deposition fraction (nDF) and the mass deposition fraction (mDF) were used, defined in Eqs. (1) and (2), respectively, as:

$$nDF = \frac{n_{Stick} + n_{Scape} \times ICRP_{correction}}{n_{Total}} \qquad (1)$$

$$mDF = \frac{m_{Stick} + m_{Scape}}{m_{Total}} \qquad (2)$$

In equation (1), $n_{Stick}$ represents the number of particles that stuck (i.e., deposit without removal or re-suspension) to the walls of the respiratory

tract, $n_{Scape}$ represents the number of particles that escape deposition, $ICRP_{correction}$ is a correction factor from the ICRP accounting for the alveolar region not modeled in the defined geometry, and $n_{Total}$ is the total number of particles in the CFPD simulation. While $m_{Stick}$ in equation (2) represents the mass of particles that stuck to the walls of the respiratory tract, $m_{Scape}$ represents the mass of particles that escape deposition, and $m_{Total}$ is the total mass of particles.

For particles deposited in the left and right lobes, in terms of mass and number of particles, the regional mass deposition fraction ($mDF_{(L, R)}$) and the regional particle deposition fraction ($nDF_{(L, R)}$) were defined as follows:

$$mDF_{(L,R)} = \frac{m_{Stick_{(L,R)}}}{m_{Total_{(L+R)}}} \qquad (3)$$

$$nDF_{(L,R)} = \frac{n_{Stick_{(L,R)}}}{n_{Total_{(L+R)}}} \qquad (4)$$

where $m_{\text{Stick}_{(L,R)}}$ and $n_{\text{Stick}_{(L,R)}}$ represent the mass and number of particles that stick to the walls of the left ($L$) and right ($R$) lobes, respectively. The $m_{\text{Total}_{(L+R)}}$ and $n_{\text{Total}_{(L+R)}}$ represent the total mass and number of particles in both lobes. The ICRP correction accounts for the deposition in the smaller bronchioles and alveolar regions that are not modeled directly in the presented simulations.

Visualization of particle distribution profiles across three of the eight computed HRT geometries is presented in Fig. 7, illustrating how small geometric variations from different HRTs can substantially influence both the deposition profile patterns and particle deposition efficiency. Minor variations in the tracheal region contributed to the increased deposition of larger particles through impaction. Consequently, the greater the geometry deviation from an idealized model, the more notable the discrepancy between mass and particle deposition fractions.

In simulations of the full HRT with uniform 10 μm particle distribution, over 90% of particles were trapped in the oral cavity and sinuses, as seen in Fig. 8. This finding is consistent with the results of Kelly J.T. et al.[58], who observed similar deposition patterns in nasal replicas. Additionally, Shi H. et al.[35] reported an 82.5% deposition efficiency for 10 μm particles at a 20 L min$^{-1}$ flow rate, further supporting these results. The higher deposition efficiency in this study is due to the higher flow rate employed, enhancing particle capture.

Furthermore, a transient analysis was conducted to assess the variability in the particle size distribution across all the geometries. The probability density was calculated at the end of the inhalation and exhalation phases when a log-normal particle size distribution was employed as input, as elaborated in the Methods section. For this analysis, snapshots of the system were taken at $t = 1$ s (end of inhalation) and $t = 2$ s (end of exhalation). Based on these snapshots, a normalized histogram with 19 bins ranging from 0 μm to 10 μm was constructed to assess each geometry's particle diameter density distribution in the lower HRT. In Fig. 9, three plots are displayed; the one on the left represents all the snapshots analyzed together, and then the original particle distribution is plotted on top of the histograms. The center and right plots depict the peaks corresponding to all geometries during the inhalation and exhalation phases, respectively. Each point for one particular diameter bin corresponds to one geometry CFPD simulation. Notably, the histogram represents a concatenation of data from all simulations rather than an average. The line plot superimposed on the data illustrates the initial particle distribution used as input.

Finally, an analysis of the two-way particle coupling was done by analyzing the pressure drop in the simulation, demonstrating the need of this coupling between fields to obtain accurate representations of the deposition and how the particles affect the airflow behavior. In Supplementary Table 3 of the Supplementary Notes 1, results are summarized for the simulations in the 8 geometries shown here, with one simulation being only the airflow (no particles) and the other one a two-way coupled simulations with particles, as presented here in this study. Further analysis can be found in the Supplementary Notes 1, in the Airflow Structures and Particle Deposition section.

## Monte Carlo radiation transport results

This study employed the PHITS radiation transport code to perform Monte Carlo simulations with the adult male mesh-type reference computational phantom (MRCP)[11]. This work demonstrates the computational capability of the framework introduced in this work to transition from a high-fidelity CFPD particle profile distribution to a point particle source distribution within the PHITS environment. A singular distribution to illustrate the effectiveness and versatility of the computational tool is presented, where future efforts will expand this analysis to encompass a broader population. Figure 10 displays the normalized flux with the number of point sources for electrons and photons using an $^{131}$I source on the left part, the electron-only flux in the center, and the relative error in absolute units on the right panel for both fields.

As visualized in Fig. 10, the simulation employed 1,150 distinct point sources representing particle deposition locations in the HRT, executed in five separate batches. This division into separate batches was necessitated by PHITS' constraint of a maximum of 500 point sources in its multi-source module. Given that $^{131}$I exhibits multiple decay modes, each point source for one particle is counted as two sources in the input file. The radionuclide, $^{131}$I,

### Table 2 | Tabulated values of particle deposition

| Geometry | mDF (%) | nDF (%) | mDF_L (%) | mDF_R (%) | nDF_L (%) | nDF_R (%) |
|---|---|---|---|---|---|---|
| mb | 85.30 | 23.10 | 23.50 | 76.50 | 40.20 | 59.80 |
| mm | 62.70 | 40.50 | 42.50 | 57.50 | 41.10 | 58.90 |
| ms | 52.40 | 30.80 | 14.20 | 85.80 | 33.90 | 66.10 |
| bb | 48.00 | 44.90 | 24.40 | 75.60 | 49.80 | 50.20 |
| bm | 81.50 | 40.90 | 81.30 | 18.70 | 50.20 | 49.80 |
| bs | 57.00 | 46.20 | 62.20 | 37.80 | 59.20 | 40.80 |
| sb | 57.90 | 39.30 | 37.40 | 62.20 | 29.10 | 70.90 |
| sm | 62.00 | 26.10 | 34.60 | 65.40 | 46.00 | 54.00 |

In the table it was quantified the mass deposition fraction (mDF), particle deposition fraction (nDF), regional mass deposition fraction (mDF$_{(L, R)}$) and the regional particle deposition fraction (nDF$_{(L, R)}$).

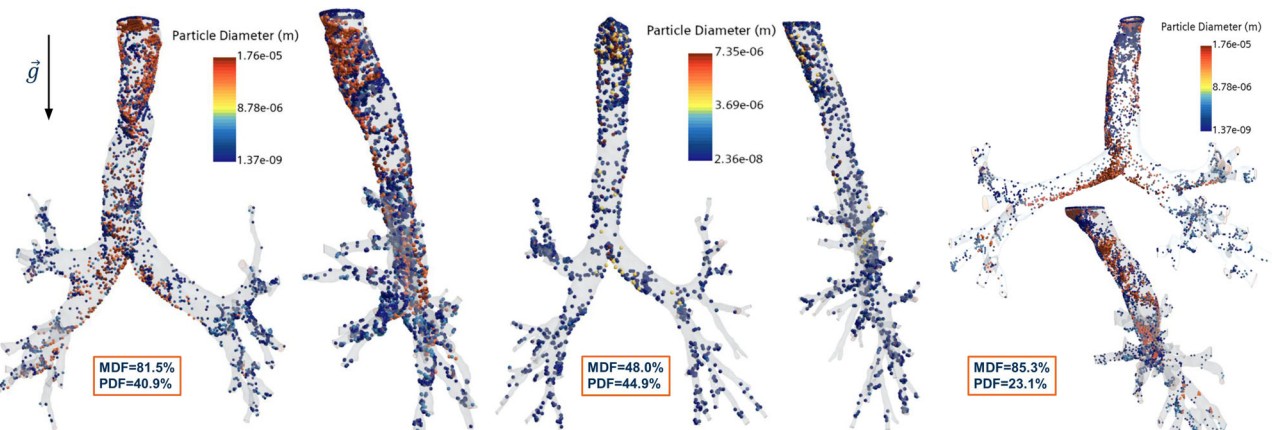

**Fig. 7 | Various particle deposition profiles at the end of the second respiratory cycle (i.e. physical time = 4 s) in the lower respiratory tract.** For each figure, the corresponding Mass Deposition Fraction (MDF) and Particle Deposition Fraction (PDF) are depicted. The PDF was corrected by the International Commission on Radiological Protection Publication 130 factors to account for the bronchi generations not explicitly modeled.

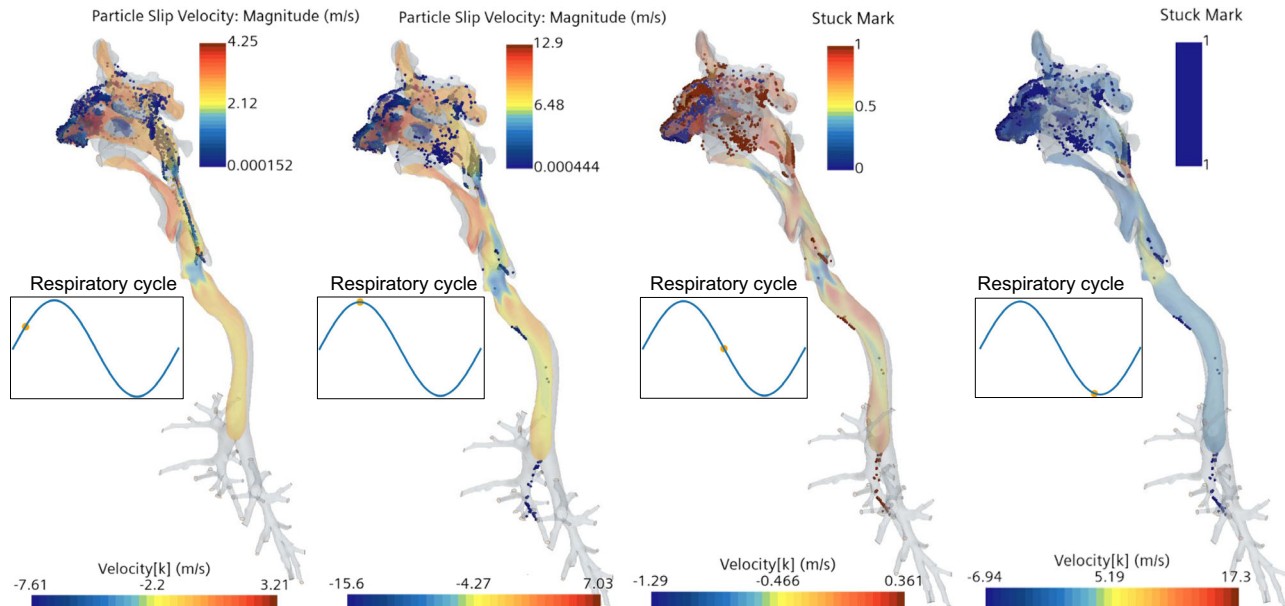

**Fig. 8 | Particle distribution patterns at different times of the respiratory cycle in the upper Human Respiratory Tract (HRT) using a uniform particle distribution of 10 μm and heavy-exercise conditions for the airflow.** The color scheme for the particles on the two figures on the left, represent the particle slip velocity while the for the two figures on the right, the Stuck Mark value was used. A value of 1 for the Stuck Mark means that the particle is stuck to the wall of the HRT, while 0 means the particle is not stuck to the wall.

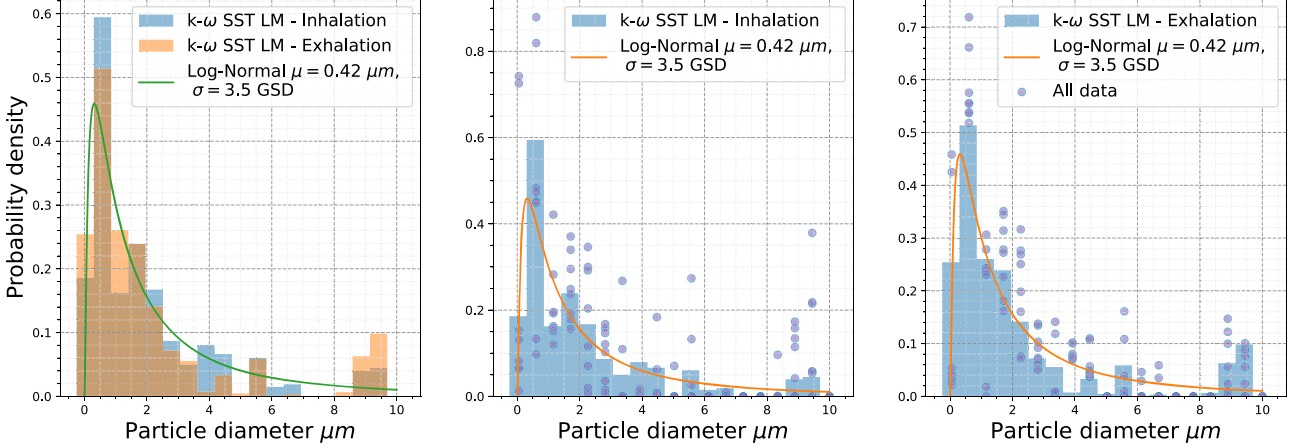

**Fig. 9 | Comprehensive particle size distribution analysis in lower human respiratory tract across all the eight geometries used in this work.** The left plot aggregates snapshots from all the geometries to display the overall particle distribution with an overlay of the original input particle distribution. The center and right plots show particle distribution peaks at the end of the inhalation (at $t = 1$ s) and exhalation (at $t = 2$ s) phases, respectively, for all geometries. Each point in the diameter bins represents data from individual CFPD simulations. The histograms displayed here are generated by concatenating all simulation data rather than averaging. The line plot superimposed on each histogram represents the original input particle distribution. The turbulence model employed in this work was the Reynolds-averaged Navier-Stokes $k - \omega$ Shear Stress Transport with the Langtry-Menter transitional flow corrections, hence the $k - \omega$ SST LM labels used for the histograms.

has a specific activity of $1 \times 10^{13}$ Bq kg$^{-1}$ for airborne particles. This activity assumes that each particle will contain 0.21% of pure $^{131}$I, based on the fact that pure $^{131}$I has a specific activity of $4.6 \times 10^{15}$ Bq kg$^{-1}$. The organ-specific absorbed dose can be found in the Supplementary Notes 1, in the PHITS Simulation section, Supplementary Table 4.

## Discussion
The first phase of this study showcased a robust and comprehensive framework for automating the reconstruction and parametrization of the HRT based on CT scans. Integrating a specialized parametrization algorithm provided valuable insights into important anatomical features that affect airflow dynamics and aerosol deposition. Features such as volume-averaged tracheal diameter and main bronchi branching angles were identified as critical parameters for predicting a subject's age and weight. The importance of these features was corroborated by employing a Random Forest regressor, thereby highlighting the tracheal diameter as the most influential parameter for both age and weight predictions.

The kernelized k-means clustering facilitated the selection of representative HRT geometries from a large database. The clusterization made CFPD simulations feasible, since it would be impossible due to time constraints and computational resources to perform CFPD simulation in the larger cohort. The strategic integration of machine learning techniques and

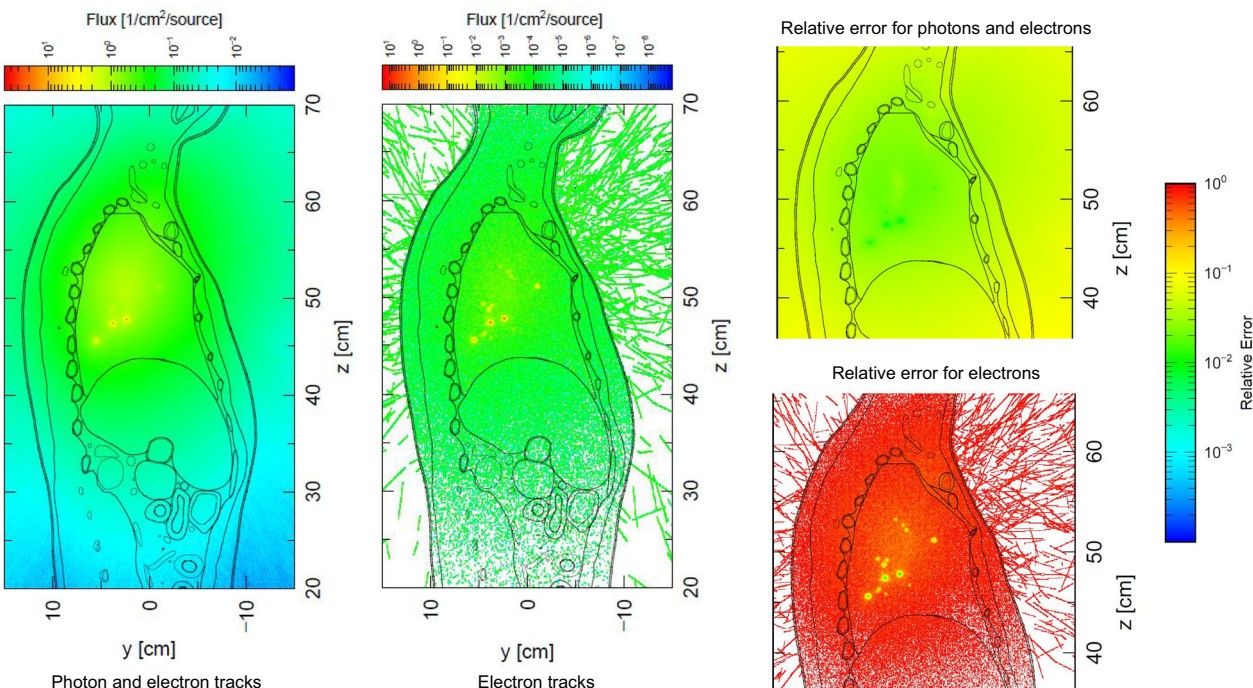

**Fig. 10 | Schematic results from the Particle and Heavy Ion Transport code System (PHITS), a Monte Carlo particle transport simulation code.** The particle deposition profiles obtained from the Computational Fluid and Particle simulations were combined with the International Commission on Radiological Protection Publication 145 computational phantoms, for a more representative distribution of radioactive particles in the lungs after an exposure event. Photon and electron flux normalized per number of sources from aerosolized $^{131}$I, with the mixed field flux shown on the figure on the left. In the center, only the electron flux is displayed. On the right, the relative errors for both the photon and electron fields are displayed.

clustering algorithms into this research computational framework refines the automated reconstruction process and adds interpretive value to the anatomical data.

The Mesh Independence studies demonstrated that one converged simulation is achieved using a base mesh size of less than 1.5 mm. All the meshes used exceeded the 2 million elements, agreeing with the criteria from prior studies as the number of elements required to constitute a converged mesh[18,20,49,59–61]. While Roache et al.[62] suggested using the number of elements as a convergence metric, it was found that using SnappyHexMesh as a meshing tool can sometimes lead to challenges in generating the desired number of elements since there is no exact user control over how many elements will be in the final mesh based on the refinement levels. Notably, the number of elements will depend highly on the geometry volume and level of detail captured. Therefore, choosing the base element size as a figure of merit is more rigorous since the final goal is to have a number that works for multiple similar geometries.

This study also reveals complex flow structures in a male population's lower and upper respiratory tract arises from small geometrical variations, notably impacting particle deposition. These results confirmed and extended the work of Fresconi et al.[63] and Kleinstreuer et al.[64], who discussed the importance of secondary motions like Dean vortices in pulmonary airways, particularly in bifurcating structures.

The observed secondary flows, characterized by tangential velocity vectors, have implications for particle deposition mechanisms, acting as forces that push particles toward the walls. The results presented in this work aligns with the results of Zhang and Kleinstreuer[60,65] and Xi and Longest[22,66], which demonstrated that secondary flows can substantially affect particle deposition. In addition, the formation of eddies in the trachea and main bronchi, although well captured by the $k - \omega$ SST LM RANS turbulence model, could not resolve smaller-scale eddies. These airflow features indicate that higher-fidelity simulation techniques, like LES, may be needed for a more comprehensive analysis, as pointed out in the study by Koullapis et al.[32].

Another notable result from this study analysis was the sensitivity of flow fields to slight geometric variations, demonstrating the need for individualized dosimetry models. Stagnation points indicated areas where diffusion, rather than impaction, dominates particle deposition in the upper sections of the lower respiratory tract. These findings aligned with the work of Shang et al.[67], who suggested that airway anatomy considerably impacts airflow dynamic patterns. Moreover, a consistent asymmetry between the left and right branches of the main bronchi was obviated, attributed in-part to the asymmetry in the lugs due to the heart location.

The inter-subject results demonstrated pronounced variability in particle deposition patterns across different HRT geometries. These results demonstrate the impact of anatomical variability in modulating particle deposition patterns, specifically in the trachea and main bronchi regions. The mDF and nDF from the CFPD models demonstrated that the extent of particle deposition was not uniformly distributed across all HRT geometries. These findings followed similar trends as earlier work by Feng et al.[68], which also reported variability in particle deposition due to anatomical differences. However, no discernible trend was identified correlating particle deposition with the selected features of trachea diameter and bronchi angle.

This study further explored the differences in the deposition profile between the right and left lung lobes, as determined by the simulated mass and particle deposition fractions ($mDF_L$, $mDF_R$, $nDF_L$, and $nDF_R$). Notably, asymmetry in deposition was observed across all geometries, where simulation results supported results from previous studies that asymmetric bifurcations in bronchial trees could lead to uneven particle distribution[69,70]. Interestingly, geometry corresponding to the bm` cluster showed a departure with 81.30% of mass deposition in the left lobe and 18.70% in the right, which suggests further investigation for under what circumstances this may happen.

Figure 7 illustrates how small changes in tracheal geometry can result in serious shifts in particle deposition patterns, reinforcing the role of impaction as a phenomenon markedly influenced by tracheal geometry. These results aligned with prior studies by Jayaraju S. et al.[71], who further

agreed that tracheal geometry influenced particle deposition. Moreover, in the transient analysis, the histograms generated by concatenating data from multiple simulations, as given Fig. 9, demonstrated how deposition profiles will follow the initial particle size distribution for the respiratory cycle. However, an increase in the particle distribution was observed for larger particles (greater than 8 μm) at the end of the exhalation, entailing that, relative to the original distribution, more particles were deposited due to impaction.

In further assessing the role of particle size on deposition in the upper HRT, results demonstrated that for particles with a size of 10 μm, more than 90% were deposited in the oral cavity and sinuses, thus indicating that for larger particles, the upper respiratory tract serves as a filter for larger particles, a phenomenon previously modeled and corroborated in literature[72,73].

The presented research findings demonstrated the impact of variability in particle deposition patterns across different HRT geometries. The study underscores the critical role of anatomical differences in determining particle deposition outcomes. It highlights the necessity, and through this study workflow, feasibility for movement towards individualized approaches in therapeutic interventions and inhaler designs. Moreover, analyzing the pressure drop in the systems with and without particles highlights the importance of considering a two-way coupled approach for the particle's dynamics integration with the airflow, since it will influence the airflow characteristics, as given in Supplementary Table 3, with further results can be given in the Supplementary Information, specifically in the Supplementary Methods, Airflow structures and particle deposition section. This result was also highlighted by Feng, Y. and Kleinstreuer, C.[74], where differences in the particle deposition efficiency was quantified in simplified models of the HRT between one-way and two-way coupling.

While the study presented here improves upon the current approaches and modeling capabilities incorporating aerosol particle size distribution, calculation of deposition, and estimation of internal dose, the proposed workflow requires consideration of the assumptions and limitations of the study. First, a singular geometry was utilized to demonstrate the capabilities of the workflow for dosimetric studies, which may not cover the full range of physiological variations and will be further addressed in forthcoming work. An expanded effort would also consider an initial particle size as a distribution of aerosolized (radioactive) particles while further considering dynamic breathing patterns for different exercise levels.

In evaluating the computational complexity of this work, using OpenFOAM, preliminary study results identified a 1–3 day time frame was required to compute the particle deposition from an entire respiratory cycle using a converged mesh and 120 cores (Dual Intel Xeon Gold 6226 CPUs @ 2.7 GHz) distributed in five computational nodes. When StarCCM+ was employed, the computational burden was reduced to 8 or 24 hours using the same number of cores. For this reason, the results shown in this research were generated entirely using StarCCM+; however, CFPD simulations for three distinct geometries were done in OpenFOAM, which can be requested by users of this solver.

## Conclusion

In this study, a semi-automated pipeline was developed that integrates CT scans to subject-specific particle deposition in the HRT and subject-specific absorbed dose assessment. Considering licensed and open-source software, several Python and Java scripts were made to transition from a CT scan to a CFPD simulation quickly. That includes, automatic HRT reconstruction from a CT scan, pre-processing process to make the 3D HRT suitable for a CFPD simulation, generation of input files for OpenFOAM and StarCCM+ solvers, and the generation of PHITS input files from a CFPD simulation results.

The primary limitation in the proposed workflow is pre-processing the geometries to be suitable for a CFPD simulation; nonetheless, the process was heavily expedited, from several hours to up to 40 minutes, depending on the user's experience. As the Monte Carlo radiation transport code employed to simulate absorbed dose profiles, PHITS software was demonstrated to be a suitable choice despite its limitations with the

maximum number of sources per run, enabling an integrated multiphysics workflow.

The clustering and parametrization results have demonstrated a strong correlation between the trachea average diameter and the main bronchi angle to predict age and weight. Although the initial objective was to correlate these anatomical markers with particle deposition patterns in the lungs, study results demonstrated that even minor perturbations in HRT geometry yielded a marked impact on mass and particle deposition fractions. The insight provided in this study mitigates the assumed primacy of tracheal diameter and main bronchi angle in governing particle deposition.

The notable impact of the methodology and results presented herein demonstrate the impact of particle deposition profiles for a diverse range of lower HRTs representing a large population. This study further demonstrates how airflow and particle dynamics differ across individual anatomies, a topic previously unaddressed. Differences as high as 30% in the mDF were seen across different HRTs, and more pronounced discrepancies were observed between the left and right lung deposition fractions, where differences exceeded 60% between lobar regions. The difference between deposition fractions will be the subject of future efforts, expanding on the results of this study. The presented work also underscores the necessity for particle coupling with airflow, as demonstrated in this research by the consequential differences between one-way and two-way coupling mechanisms. Previous investigations[17–20,22,32,33,35,48,49,66] have often limited their scope to one-way coupling, overlooking the complexity revealed by the more rigorous approach chosen in this work, and light-exercise respiratory breathing conditions.

Regarding the PHITS simulations, a tool to perform the runs starting from the particle track file that StarCCM+ will output is presented and successfully implemented. Nonetheless, this tool could be expanded to encompass OpenFOAM particle track outputs. The results from PHITS simulations showed flux hotspots that would otherwise not be captured if assuming a uniform radioactive activity distribution in the lungs, which impacts the calculation of the mechanism and etiology of radiation-induced health effects from internalized particulates.

The present study furnishes a comprehensive framework for subject-specific internal dosimetry, laying the foundation for the next generation of dosimetry tools. This tool was successfully implemented in many different HRT geometries, showcasing the importance of different features within the HRT. Results on the performed studies are limited to only one particle distribution resulting from aerosolized [131]I released to the environment, heavy exerciser breathing conditions, and a representative respiratory cycle.

Ongoing efforts are focusing on improving the algorithms presented to fully automatize the process of going from a CT scan to a CFD or CFPD simulation, which will further reduce the time spent on pre-processing to a timescale in minutes. Moreover, future efforts focus on employing different initial source term particle distributions, accounting for electrostatic effects on the particle forces and the lower HRT used in the ICRP Publication 145 mesh phantoms. One limitation is that the adult male phantom does not have a detailed HRT defined. Therefore, a more accurate mapping of the HRT is required for the ICRP Publication 145 phantom to account for realistic HRTs.

## Methods

For this study to be feasible the a key innovation was to develop an automatized workflow for a fast transition from a CT scan to personalized radiation dosimetry based on high-fidelity particle distribution profiles using CFPD. Using the particle distribution from the CFPD simulation in the MRCP computational phantom developed by Kim C.H. et al.[11], subject-specific and high-fidelity dose maps were obtained. The workflow was based on four main steps. A visual summary of the methodology is pictured in Fig. 11, and it can be summarized as follows:

1. Automated Python algorithms to obtain 3D reconstructions of the lower and upper HRT using CT scans.
2. 3D Geometry pre-processing using semi-automated python scripts in Blender[46] to pre-process the geometry to be suitable for a CFPD simulation.

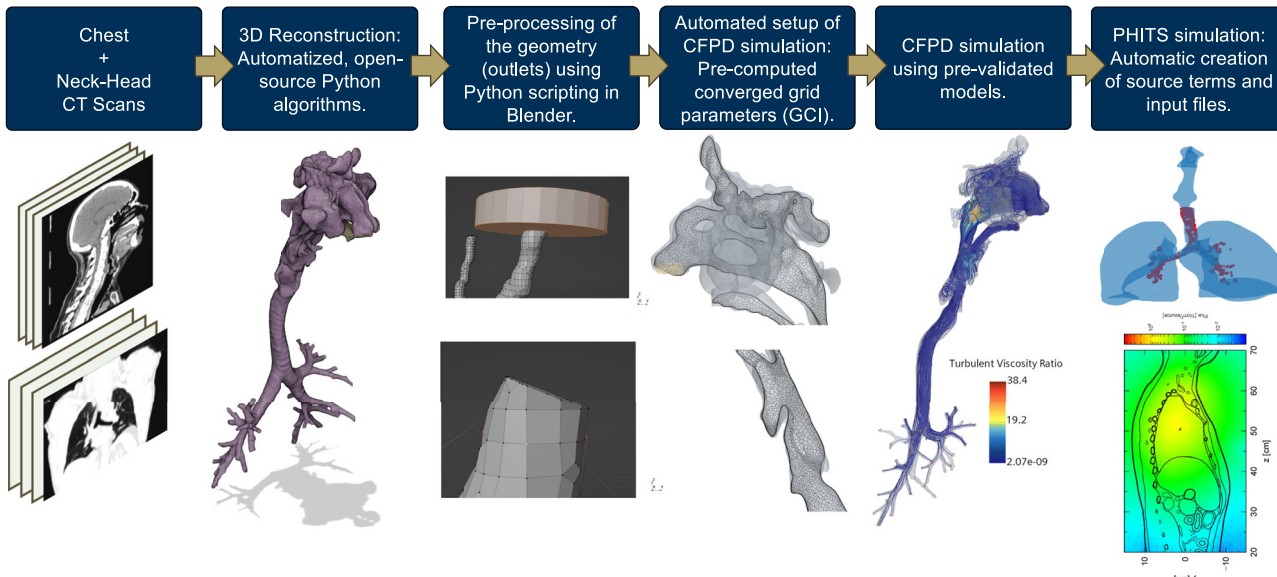

**Fig. 11 | Graphical representation of the hybrid-automatic workflow to obtain personalized radiation dosimetry starting from a Computed Tomography (CT) scan and using Computational Fluid and Particle Dynamics (CFPD).** The figure illustrates the workflow for obtaining 3D reconstructions of the lower and upper human respiratory tract (HRT) using CT scans through automated Python algorithms. The 3D geometries are pre-processed using semi-automated Python scripts in Blender to make them suitable for CFPD simulations. Automated processes then prepare files required for converged CFPD simulations under realistic breathing conditions, supporting both open-source software (OpenFOAM) and licensed software (StarCCM+). The CFPD simulations employ Reynolds-Averaged Navier-Stokes (RANS) equations for airflow and utilize two- and four-way coupled Lagrangian tracking for particles, depending on the solver (MPPICFoam in Open-FOAM and the DPM Method in StarCCM+). Finally, the workflow includes generating source and input files for PHITS one-way coupling and running Monte Carlo simulations with the MRCP adult phantom.

3. Automatic preparation of the necessary files for a converged CFPD simulation using representative source terms and realistic breathing conditions. Both for Open-Source software (OpenFOAM[41]) and licensed software (StarCCM+[42,75]).

4. CFPD simulation using RANS equations for the airflow and a two and four-way coupled Lagrangian tracking for particles depending on the solver (MPPICFoam in OpenFOAM and DPM Method in StarCCM+).

5. Source generator and input file generator for PHITS one-way coupling and run the Monte Carlo simulation using the recently developed MRCP adult phantom[11].

In the following subsection, each step will be explained in detail.

**Geometry reconstruction**

The first step of this algorithm involved the reconstruction of the geometry and subsequently identifying representative clusters from the population that define multiple geometries that account for the diversity in the database obtained for the case of the lower HRT. The CT scans from this study were obtained from various publicly available databases, as noted below. The average voxel dimensions, the standard deviation, and minimum and maximum values are given in Supplementary Table 5 of the Supplementary Methods. For the lower respiratory tract, the following databases were used:

- From the EXACT'09 challenge[25] 46 de-identified chest CT scans were obtained (From the trachea up to the 5th to 7th generation of bronchi depending on the quality of the CT scan). The conditions of the scanned subjects varied widely, from healthy volunteers to subjects with severe abnormalities in the airways or lung parenchyma.
- Pediatric Chest/Abdomen/Pelvic CT Exams with Expert Organ Contours (Pediatric-CT-SEG)[76,77] from The Cancer Imaging Archive (TCIA)[78] represent 359 random pediatric cases based upon routine clinical indications. No medical or diagnostic data are available for any subject dataset.
- A Large-Scale CT and PET/CT Dataset for Lung Cancer Diagnosis (Lung-PET-CT-Dx)[79] from TCIA[78]. All the 355 subjects in the database

were diagnosed with Adenocarcinoma, Small Cell Carcinoma, Large Cell Carcinoma, and Squamous Cell Carcinoma.

- Chest Imaging with Clinical and Genomic Correlates Representing a Rural COVID-19 Positive Population (COVID-19-AR)[80–82] from TCIA[78] includes data collected from 105 hospitalized subjects with a positive COVID-19 laboratory test verified diagnosis, with imaging studies conducted within eight days prior to diagnosis and at least one imaging study post-diagnosis.
- CT Ventilation as a Functional Imaging Modality for Lung Cancer Radiotherapy (CT vs PET Ventilation Imaging)[83,84] from TCIA[78] includes health information of 16 subjects all of whom had mild, moderate, or severe chronic obstructive pulmonary disease and impairment in the diffusing capacity of the lungs for carbon monoxide.

To investigate the upper respiratory tract—which encompasses the trachea, pharynx, oral cavity, and nasal cavity— 111 de-identified head-neck CT scans from the TCIA archive[78] were acquired, specifically sourced from the Head-Neck Cetuximab database[85,86].

Using the mentioned CT scan datasets, the next step was to reconstruct the 3D geometry of the upper HRT. Head-neck CT scans were required to be processed separately from chest CT scans to obtain the upper and lower HRT geometries, respectively. The upper HRT reconstruction step was done by adapting and updating the algorithm by Cercos-Pita et al.[87]. In Supplementary Methods, Geometry Pre-processing section, it can be found the specifics parameters and modifications to the original implementation.

For the lower respiratory tract, a 3D-UNet architecture was employed, based on the NaviAirway framework[44]. Post-processing tools were adapted from the airway extraction methodologies detailed in the work of Garcia-Uceda et al.[45]. The neural network was trained on the widely used EXACT'09 training dataset[25] and subsequently validated it on a separate dataset. Following the 3D geometric reconstruction of the lower respiratory tract, automated parametrization was executed to quantify key metrics, including trachea length, average trachea diameter, G0-to-G1 branching angle, and overall reconstruction volume.

This parametrization provided a heterogeneous mix of phenotypical and non-phenotypical metrics characterizing the CT scan database for the lower HRT. Random Forest regression was applied to identify the most predictive features of the lower HRT for estimating a subject's age and weight. The final preparatory step for CFD/CFPD simulations involved smoothing the 3D geometry and defining inlets/outlets. Further explanation of the pre-processing algorithm is given in Supplementary Methods, Geometry Pre-Processing subsection.

## CFPD modeling

Using an automated meshing process, a RANS approach is selected to model the turbulence. In particular, the $k - \omega$ SST LM model was selected[88]. This model was validated for particle deposition by Koulappis et al.[32] against LES simulations in the upper and part of the lower HRT, which was also employed by Zhang and Kleinstreuer[60], as well as Shang. et al[67]. The boundary conditions for pressure, turbulent kinetic energy ($k$), and turbulent specific dissipation rate ($\omega$) were rigorously defined as these conditions would most remarkably impact the model convergence and final results.

The boundary conditions for the velocity $\boldsymbol{u}$ were chosen to replicate a representative respiratory cycle of a human subject under heavy-exercise breathing conditions. To replicate these conditions, the inlet flow was represented as $Q(t) = Q_{\max} \sin\left(\frac{2\pi t}{T}\right)$, please refer to Supplementary Fig. 5 in Supplementary Methods to see the details of this representation. Supplementary Fig. 5 of the Supplemental Methods in the Supplementary Information further details realistic heavy-breathing condition pattern from the average breathing pattern of a human subject. For the meshing process, the hexahedral-based meshes for OpenFOAM and polyhedral meshes for StarCCM+ were employed in this study. To perform the meshing process in OpenFOAM, the preferred tool was `SnappyHexMesh`, while in StarCCM+, the polyhedral volumetric mesh was prefered. The difference between a hexahedron-dominant mesh and a polyhedral mesh did not affect the PDPs, as discussed by Thomas, M. L. and Longest, P. W.[89], demonstrating no significant differences in deposition efficiency values using a simplified bifurcation geometry. An automatized script was developed to perform the meshing and to perform the runs of the simulation in Python when using OpenFOAM, while using StarCCM+ macros, an automatized Java script was also developed to automatize the simulations in the aforementioned licensed software.

In the OpenFOAM implementation, the `SnappyHexMesh` tool generated the body-fitted mesh. Various parameters were tested to propagate the boundary layer well through the geometry to achieve this efficiently and maintain the mesh quality during the process. It was further identified that a well-posed background mesh, where the elements conform a cuboid geometry, will avoid high skewness later. To achieve such background mesh, the 3D HRT is bounded in a box, and then the box is divided an integer number of times to fit cubic elements. The size of the cubic element is taken according to the size of the minor outlet, therefore having at least one element intersecting each surface, as recommended by the `Snappy-HexMesh` user manual. If the size of the minor outlet exceeds 1.5 mm, we set the cubic element to be 1.5 mm, based on our mesh independence studies. In StarCCM+, the element base size is a global parameter that sets the initial tetrahedral cell size before any refinement to a polyhedral mesh and local sizing controls are applied.

Particle-flow coupling employed a four-way coupled Lagrangian approach in OpenFOAM using the Multi-Phase Particle In Cell (MPPIC) solver. In contrast, in StarCCM+, a two-way coupled Lagrangian tracing was opted for, using a Lagrangian Multi-phase (LMP) solver. MPPIC and LMP solvers account for drag, lift, virtual mass and gravity forces. However, in the StarCCM+ LMP solver, an additional force was included using the user-defined volumetric forces for accounting for the Brownian diffusion for nano-sized particles.

Equation (5) defines the transport of particles, where $C_d$ is the drag coefficient of the particle, which is dependent on the implementation of the drag model used; in this work, the Schiller-Naumman correlation was used since it is the only drag model available for solid particles in StarCCM+. It

should be noted that the drag model used in this work does not consider the non-continuum slip effects on particles at the submicron level, which can be corrected by using the Cunningham slip correction factor. The airflow velocity is represented by $\mathbf{u}$, and $\alpha_p$ is the volume fraction of the particles in the air, which in this case will be close to zero. The mass of the particle is denoted by $m_p$.

$$m_p \frac{\partial \mathbf{v}_p}{\partial t} = \frac{1}{2} C_d \rho A_p |\mathbf{v}_s| \mathbf{v}_s + m_p \mathbf{g}(1 - \alpha_p) + \mathbf{f}_{\text{Brownian}} + \mathbf{f}_{\text{lift}} + \mathbf{f}_{\text{virtualmass}}$$

(5)

where $C_d$ is the drag coefficient of the particle, $\rho$ is the density of the continuous phase, $\mathbf{v}_s = \mathbf{u} - \mathbf{v}_p$ is the particle slip velocity, with $\mathbf{u}$ being the instantaneous velocity of the continuous phase. The $A_p$ is the projected area of the particle, $\mathbf{g}$ is the gravitational acceleration, $\mathbf{f}_{\text{Brownian}}$ is the force due to Brownian motion, $\mathbf{f}_{\text{lift}}$ is the lift force, and $\mathbf{f}_{\text{virtual mass}}$ is the virtual mass force.

MPPIC solver does not explicitly resolve particle-particle interaction since this will be computationally expensive, and given that the particle packing fraction is less than $10^{-3}$, there was no identified need to account for explicit particle-particle interactions. Nonetheless, in the latest OpenFOAM implementations (since 2014), the particle-particle interactions are represented by models that utilize mean values calculated on the Eulerian mesh[90]. In this study, this is called a four-way coupling since particle-particle collision is considered but has yet to be explicitly modeled.

MPPIC and LMP solvers account for drag, lift, and gravity forces. However, in the StarCCM+ LMP solver, an additional force was included using the user-defined volumetric forces for accounting for the Brownian diffusion for nano-sized particles. The Brownian force $\mathbf{F}_B$ expression for a particle in a fluid medium as defined by Li and Ahmadi[91] is expressed in equation (6) and can be used for both laminar and turbulent flow regimes.

$$\mathbf{F}_B = \sqrt{\frac{2k_b T_o \alpha}{\Delta t}} \boldsymbol{\xi}$$

(6)

Where $k_b$ is the Boltzmann constant, $T_o$ is the temperature of the fluid in Kelvin, $\alpha$ is the particle's mobility, $\Delta t$ is the time step, and $\boldsymbol{\xi}$ is a vector of Gaussian random numbers with zero mean and unit variance that will define the direction of the force. The particle mobility $\alpha$ is a measure of how easily a particle moves when subjected to a force and is defined as $\alpha = \frac{D}{kT}$ where $D$ is the diffusion coefficient. The diffusion coefficient $D$ can be calculated using the Einstein relation for spherical particles: $D = \frac{kT}{3\pi\eta d}$, where $\eta$ is the air viscosity and $d$ is the particle diameter.

The boundary conditions for the pressure are fixed value at the outlets, pressure gradient equal to zero at walls, and at inlets. For the turbulent kinetic energy $k$ in the inlets, it can be defined assuming isotropic turbulence as $k = \frac{3}{2}(I||\mathbf{u}_{\text{ref}}||)^2$. The turbulent intensity was set equal to $I = 0.04$, considering that our simulations will be a medium-turbulence case with 4% of turbulence intensity, the $\mathbf{u}_{\text{ref}}$ is taken to be the peak of the velocity in the inhalation, which will be geometry dependent since the fixed magnitude is the flow. As for the turbulent specific dissipation rate flow $\omega$ in the inlet it was computed as $\omega = \frac{k^{0.5}}{C_\mu^{0.25}L}$, where $C_\mu^{0.25}$ is a constant equal to 0.09, and $L$ is a reference length scale, that was taken to be the hydraulic diameter $d_h$ of the inlet or inlet(s) in the case of nasal inhalation. The outlets' $k$ and $\omega$ condition is the Neumann boundary condition equal to zero. While in StarCCM+, the wall treatment is done by activating the -All y+ treatment- feature, in OpenFOAM, this requires manual handling. For the turbulent kinetic energy $k$, the `kLowReWallFunction` was used, which can handle both low and high-Reynolds regimes. For $\omega$ instead, the standard `omega-WallFunction` was used. This combination of wall functions is the most stable for the family of SST transitional models.

To avoid any confusion, the `kLowReWallFunction` boundary condition provides a wall constraint on the turbulent kinetic energy for low- and high-Reynolds number turbulence models, as per the user manual. Additionally, using a low Re model for $k$ is critical to the model due to the

**Table 3 | List of models, parameters, and boundary conditions for the airflow and particles in the simulations presented in this work**

| Description | Parameter Value/condition (Flow Solver) | Description | Parameter value/condition (flow solver) |
|---|---|---|---|
| Turbulence model | $k - \omega$ SST LM | Wall condition for $\vec{u}$ | No slip boundary condition |
| Inlet boundary conditions for $\vec{u}$ | Variable flow rate: $Q = Q_{max} \sin(\frac{2\pi t}{T})$ | Airflow density and temperature | 1.177 kg/m³; 300 K |
| Outlet boundary conditions $\vec{u}$ | Zero gradient | Particle solver | MPPICFoam (OpenFOAM); Lagrangian Multi Phase (StarCCM+) |
| Inlet boundary conditions for $p$ | Zero gradient | Particle coupling | MPPICFoam (OpenFOAM); Lagrangian |
| Outlet boundary conditions for $p$ | Fixed pressure value (static pressure of the environment) | Particle distribution density function: | log-normal, $\mu = 0.42$ µm, and $\sigma = 3.5 \frac{1}{x\sigma\sqrt{2\pi}} \exp\left(-\frac{(\ln x - \mu)^2}{2\sigma^2}\right)$ |
| Inlet boundary conditions for $k$ | $\frac{3}{2}(I(\|u_{ref}\|))^2)$ $I$=0.04, 4% turbulence intensity | Particle density | iodine, $\rho = 4390$ kg/m³ |
| Inlet boundary conditions for $\omega$ | $\omega = \frac{k^{0.5}}{C_\mu^{0.25}L}$ | Number of particles injected to the system | 100,000 particles (OpenFOAM) |
| Inlet boundary conditions for $\gamma$ | $\gamma = 1$ | Injection rate | 1/(time step) per injector point (StarCCM+) |
| Inlet boundary conditions for $Re_\theta$ | See Supplemental Information, CFPD modeling | Particle-Wall interaction mode | Stick upon contact with the wall |
| Outlet boundary conditions for $\gamma, Re_\theta, k, \omega$ | Zero gradient | Time integration scheme | PISO algorithm (OpenFOAM); Implicit unsteady (StarCCM+) |
| Wall treatment for $k$ | All $y^+$ treatment (StarCCM+); kLowReWallFunction (OpenFOAM) | Time step used | 0.015–0.025 s |
| Wall treament for $\omega$ | All $y^+$ treatment (StarCCM+); omegaWallFunction (OpenFOAM) | Total time simulated | 4 s. (2 s. flow only and 2 s. with particles) |

A more detailed explanation of the selection of these parameters can be found in the Supplemental Information, CFPD Modeling section.

intermittency parameter. The fluid in our study transitions between turbulent and laminar regimes, and a model that accounts for this transition without numerical instability is needed. Near the walls, at low Reynolds numbers, the turbulence kinetic energy can exhibit problematic behavior if not adequately modeled, leading to inaccuracies or solver failures.

The Langtry-Menter model introduces additional intermittency equations ($\gamma$) and transition momentum thickness Reynolds number ($Re_\theta$). For these equations, the boundary conditions in the inlet are $\gamma = 1$ and for $Re_\theta$ the definition is more complex, and it is defined in equation (7), where $Tu = 100 \frac{\sqrt{2/3k}}{\|\mathbf{u}_{ref}\|}$. For the walls and the outlets, the Neumann boundary conditions were applied ($\frac{\partial}{\partial n}\gamma = 0, \frac{\partial}{\partial n}Re_\theta = 0$).

$$Re_\theta = \begin{cases} 1173.51 - 589.428Tu + \frac{0.2196}{Tu^2} & \text{if } Tu \leq 1.3 \\ \frac{331.5}{(Tu-0.5658)^{0.671}} & \text{if } Tu > 1.3 \end{cases} \quad (7)$$

The particle size distribution was modeled as a log-normal distribution with a mean size of 0.42 µm and a geometric standard deviation of $\sigma = 3.5$. This choice was informed by the known characteristics of aerosolized $^{131}$I particles released during the Chornobyl NPP incident[6,37]. In MPPICFoam simulations, a fixed total of $1 \times 10^5$ particles were injected. Conversely, in simulations employing StarCCM+ LMP, the number of particles injected was geometry dependent, with a constraint of one particle per parcel to maximize the accuracy of particle-airflow interactions. Employing a fixed particle injection rate could lead to parcels containing an excessive number of particles, thus diminishing the model's accuracy and necessitating the injection of additional parcels to achieve adequate statistical representation. It is worth noting that the concept of particles and parcels within StarCCM+ is complex and requires a thorough understanding to avoid misinterpretation. The one-particle-per-parcel constraint was therefore adopted to mitigate this complexity. Importantly, even with varying numbers of particles across different simulations, none had fewer than $5 \times 10^4$ particles, ensuring sufficient statistical variance for meaningful interpretation of results.

With the turbulence models well established, the CFPD simulation setup was automated for both StarCCM+ and OpenFOAM. In OpenFOAM, a Python script initializes all necessary dictionaries, including the

velocity (**u**), turbulence kinetic energy ($k$), specific rate of dissipation ($\omega$), turbulent viscosity ($\nu_t$), pressure ($p$), and intermittency ($\gamma_{int}$), as well as Reynolds shear stress ($Re_{\theta t}$). The script also generates dictionaries required for the MPPIC solver's colliding cloud model.

In StarCCM+, a Java script was employed to prepare the simulation automatically. Upon importing the geometry and assigning the relevant patches (inlet, wall, or outlet), the script applies the pre-defined models and initial conditions, rendering the simulation ready for execution.

A summary of the models, parameters, and values used in this work is tabulated in Table 3.

**PHITS model definition**

Once the high-fidelity PDPs from the CFPD simulations were obtained, a particle distribution profile was selected to be imported into the male MRCP to obtain the dosimetric information. The mesh-type phantoms were adopted from the voxelized ICRP Publication 110[92]. The general structure, the HRT, and the lungs highlighted inside the phantom can be seen in Supplementary Fig. 6 in the PHITS modeling section of the Supplemental Methods in the Supplementary Information.

A tool was created in Python to read the track files from the CFPD simulation in StarCCM+ and properly scale and rotate the particle cloud to fit the HRT in the MRCP phantom better, as detailed in Supplementary Fig. 7 of the Supplementary Methods. A further detailed explanation can be found in the PHITS Model definition section, on Supplementary Methods.

For this study, $^{131}$I was selected as the radionuclide of interest for simulating internal dosimetry. Iodine-131 is a radionuclide with a half-life of approximately 8.02 days, predominantly decaying via both beta $\beta^-$ and gamma $\gamma$ emissions. $^{131}$I decays into stable Xenon-131 ($^{131}$Xe) through $\beta^-$ decay, accompanied by the emission of gamma radiation from the $^{131}$Xe excited states. The beta particles have a maximum energy of 0.606 MeV (89.6%), 0.334 MeV (7.34%), and 0.807 MeV (0.39%), while the primary gamma emission has an energy of 0.364 MeV (81.5%)[93].

Iodine-131 is a volatile radionuclide that poses serious risks when inhaled, as it accumulates in the thyroid gland. Its $\beta^-$ emissions are particularly damaging to soft tissues and can increase the risk of thyroid cancer and other thyroid-related diseases[94]. In particular, $^{131}$I is regarded as one radionuclide of interest because of its high fission yield and high radiation

risk to humans, especially children. Due to its dual decay modes, [131]I provides a more comprehensive simulation environment than cesium-137 ([137]Cs). Cesium-137 primarily decays via gamma emission and lacks beta decay, contributing to the local biological hazard posed by [131]I. The dual decay modes allow this study to encompass the greater complexity of inhaled radionuclides. Moreover, [131]I is a common decay product in the fission of uranium and plutonium, making it prevalent in nuclear fallout[94]. Its volatility and ability to spread make it essential to account for in models and simulations related to internal dosimetry in fallout scenarios[6,95].

A Python script was done to automatize creating the input files for PHITS, using the MRCP phantom and a given track file for the particles from StarCCM+.

### Reporting summary

Further information on research design is available in the Nature Portfolio Reporting Summary linked to this article.

### Data availability

A permanent repository for the geometries used in this work is available at https://doi.org/10.6084/m9.figshare.24787773[96]. All the final particle deposition profiles and airflow datasets are available from the corresponding author upon reasonable request but not published due to the large size of each CFPD simulation dataset. The following databases were used from publicly available databases. For the lower respiratory tract: 1. EXACT'09 challenge[25]. 2. Pediatric Chest/Abdomen/Pelvic CT Exams with Expert Organ Contours (Pediatric-CT-SEG)[76–78]. 3. A Large-Scale CT and PET/CT Dataset for Lung Cancer Diagnosis (Lung-PET-CT-Dx)[78,79]. 4. Chest Imaging with Clinical and Genomic Correlates Representing a Rural COVID-19 Positive Population (COVID-19-AR)[78,80–82]. 5. CT Ventilation as a Functional Imaging Modality for Lung Cancer Radiotherapy (CT vs PET Ventilation Imaging)[78,83,84]. For the upper respiratory tract, 111 de-identified head-neck CT scans from the Head-Neck Cetuximab database[78,85,86] were used.

### Code availability

Access to the (GitHub) repository with the code used in this work can be granted for academic or research purposes upon reasonable request to the corresponding author and under appropriate confidentiality agreements. Requests will be evaluated on a case-by-case basis.

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

## Acknowledgements

Funding for this project is provided by the Department of Defense - Peer Reviewed Medical Research Program (DOD PRMRP) under award number W81XWH-21-1-0984 (S.A.D., M.T.R., I.R.B.) and National Institutes of Health/National Institute of Allergy and Infectious Diseases (NIH/NIAID) Radiation and Nuclear Countermeasures Program under award P01AI165380 (S.A.D., M.E.T., and M.S.G.P.). This research was supported in part through research cyberinfrastructure resources and services provided by the Partnership for an Advanced Computing Environment (PACE) at the Georgia Institute of Technology, Atlanta, Georgia, USA. This research made use of the resources of the High Performance Computing Center at Idaho National Laboratory, which is supported by the Office of Nuclear Energy of the U.S. Department of Energy and the Nuclear Science User Facilities under Contract No. DE-AC07-05ID14517.

## Author contributions

I.R.B. conducted the technical work and wrote the manuscript; M.S.G.P. helped with the technical work in the reconstruction and clusterization process; M.E.T.R. conceived and led the experiment and reviewed the manuscript; S.D. conceived the experiment and reviewed the manuscript.

## Ethics declaration

This study involves the use of CT scans to reconstruct HRTs geometries for computational simulations. All CT scans used in this research were obtained from publicly available, anonymized datasets or through proper agreements from relevant institutions, ensuring no direct identification or personal data of individuals were included. This research follows the guidelines set forth by the Declaration of Helsinki and we consulted local institutional review boards (IRBs) to confirm that this research is exempt of IRB. No human subjects were recruited or involved in any way during the computational experiments, and the personalized modeling approach does not involve any direct interventions on human participants. All simulations were conducted in-silico using anonymized and de-identified data available online.

## Competing interests

The authors declare no competing interests.
