## [Peer Review File · Communications Engineering]

Computational Multiphysics Modeling of Radioactive Aerosol Deposition in Diverse Human Respiratory Tract Geometries

Corresponding Author: Dr Shaheen Dewji

Version 0:

Reviewer comments:

Reviewer #1

(Remarks to the Author)

In this paper, the authors introduce a comprehensive computational framework to generate patient-specific particle deposition profiles and radiation dose maps from CT scans, using computational fluid-particle dynamics (CFPD) simulations and Monte Carlo radiation transport. The study analyzed a large database of 3D human respiratory tract (HRT) geometries reconstructed from CT scans, clustered them based on relevant parameters using machine learning, and performed CFPD simulations on representative geometries. The paper quantifies inter-subject variability in particle deposition patterns and airflow structures due to differences in HRT geometry. It also couples the CFPD particle deposition profiles with the PHITS Monte Carlo code to generate detailed radiation dose maps, highlighting non-uniform energy deposition, and demonstrates the importance of accounting for anatomical variations in HRT position and radiation dosimetry, going beyond the generic ICRP models.

The claims in this paper are largely novel. While previous studies have explored CFPD simulations and CT-based HRT geometry reconstructions, this is one of the first large-scale studies accounting for realistic population diversity and integrating it with radiation dosimetry. The coupling of CFPD with Monte Carlo for subject-specific dosimetry, automated geometry reconstruction, and machine learning-based clustering are all novel aspects.

The paper is of significant interest to researchers working on respiratory dosimetry, aerosol deposition, radiation protection, and related fields. The computational framework and insights into inter-subject variability can contribute to more accurate risk assessment and personalized therapeutic approaches. The paper has the potential to influence thinking in the field by highlighting the limitations of generic ICRP models and the need for subject-specific dosimetry approaches, especially for scenarios involving diverse populations exposed to radioactive aerosols.

Based on the novelty, significance, and convincingness of the claims, as well as the potential impact on the field, I would recommend that the authors be encouraged to consider a revision for addressing the potential concerns listed below. With appropriate revisions, this study could make a valuable contribution to the field of respiratory dosimetry and radiation protection.

1. For the CT scan dataset, "542 reconstructed 3D geometries of the lower human respiratory tract (HRT) from subjects aged 2 to 90 years, with 54% males and 46% females, including 88 pediatric subjects", among all the subjects, are their lungs all in healthy condition or some are diseased lungs? Would there be any other conditions that may affect the morphology of the lower HRT?
2. On page 3, it states that "Only four complete HRTs were connected using Blender", can the authors provide more details on criteria used for selecting those four HRTs? What are the upper HRTs' parameters that matter most regarding the particle deposition?
3. On page 5 line 131-141, there seems 2 redundant groups, specifically "mean td and mean ba (mm)".
4. Can the authors provide the definition of base size used in mesh independence test, and details on how it was used in the mesh generation?
5. The author explained explicitly the importance of choosing appropriate wall functions when using SST LM RANS model. Can they justify their reason for applying "low Reynolds wall function for k and a regular wall function for ω was used at the walls" in their modeling work?
6. Have the authors considered validation of particle deposition profiles and radiation dose maps against experimental data or clinical measurements for selected HRT geometries? For Figure 5, were the clinical data based on in vitro test or in silico results? Was the same lung geometry used in previous studies? If different lung model was used, how to justify the effect of HRT morphology on particle deposition pattern?
7. In Figure 5 right plot, it seems that deposition efficiency in sm is hardly affected by the impaction parameter within 10 to 1e10? Do the authors have any possible explanation regarding this finding?
8. The paper does an adequate job of discussing the claims in the context of previous literature, citing relevant studies on CFPD simulations, CT-based geometry reconstructions, and radiation dosimetry models. However, a more comprehensive discussion of previous subject-specific dosimetry efforts and their limitations could further contextualize the novelty of this

work.

Reviewer #2

(Remarks to the Author)

Advanced Patient-Specific Computational Insights into Aerosol Deposition and Radiation Exposure in Diverse Human Airways

This work highlights the importance of studying the contribution of geometrical characteristics of human respiratory tracts (HRT) that are specific to individuals on subject-oriented radiation dosimetry tools. These geometrical variances can affect the distribution of deposited radioactive aerosol upon radiological exposure on HRT walls, which is the focus of the work. In the paper, a synergistic computational approach that uses (1) the numerical methods of computational fluid dynamics (CFD) and computational fluid and particle dynamics (CFPD) on reconstructed 3D HRTs are implemented to simulate the deposition of inhaled aerosols on HRT walls. This approach is coupled with (2) Monte-Carlo simulations to generate energy deposition maps; and (3) the random forest regression learning algorithm to predict individual weight and age from reconstructed HRT geometries, identifying the most significant features in the HRT different geometries. In this work, findings support the significance of personalized dosimetry studies due to particle deposition being highly dependent on small variations of the HRT geometry. The methodology approach presented is found to be of high interest in the breath aerosol research community in general and, more specifically, drug delivery in addition to the current application. Although, the paper is relatively well organized; the level of a comprehensive computational setup and analysis is substantially limited with some incongruencies as well as inadequate in details, consistency, and clarity; particularly in the field of (1) above. Observations are in detail below and based on those, this paper is not recommended for publication in this journal under the present version.

1. This paper is presented in such a fashion that even when the journal is oriented to the engineering community, the reader must be quite familiar with the specific numerical methods used and infer important model information that are not found in either the paper or its supplementary.
 - a. The numerical methodologies of CFD and CFPD are interchanged back and forth without making any distinction between the two of those. CFD is not defined in L41 and the primary differences relevant to the field of study between these two methodologies are missing. Authors should define both methodologies and state what the advantages of CFPD are in terms of predicting particle trajectory and distribution and what the advantages of CFD are in terms of fluid dynamics and how they can be coupled.
 - b. Elaboration is needed when comparing RANS vs. LES models for HRT flow characterization. What are the implications of selecting one method over the other in terms of accuracy per HRT segment as well as computational time?
 - c. Model inputs, including fluid and particle material properties used, in an organized fashion could be useful not only clarity but also to allow reproducibility.
 - i. In L212, the particle dynamics are set as unsteady but not details of the particle time steps are provided per particle size distribution/injections. Moreover, it is not clear what the boundary condition describing the particle-wall interaction is. It could be inferred they are “trapped” based on L208 but this is unclear whether particles are trapped upon a single contact with the wall?
 - d. In L143, the authors assumed a “low Reynolds” flow for their choice of turbulence treatment on the wall, which also contradicts their argument to implement a two-way coupling approach. What would the basis be for such assumptions having a peak of 90 L/min flowrate in the domain studied? A contour-plot figure showing the fluid Re numbers throughout all HRT segments is needed to support such assumption and prevent subjectivity.
 - e. L18-19 in Appendix I: There is no clarity about what the authors want to convey by “fixed pressure boundary conditions” at the outlets.
 - f. “Element base size” or “mesh size” expressed as cell length is preferred as well as including the domain length, for comparison purposes, and a scale bar added to Figure 1 in Appendix 1.
 - i. Furthermore, in L25 in Appendix I, the basis to establish high-fidelity simulations for “base size of 0.054 mm to establish a benchmark for convergence analysis” doesn’t seem to be sufficient based merely on mesh refinement with no further information (see ii below).
 - ii. No element quality is reported, this is needed to validate model robustness. Also, a histogram per case studied showing the average of cell quality in the model should be included in Figure 1, Appendix 1.
 - iii. L27, Appendix 1: Residual value in the CFD-CFPD models should be clearly quantified to support model fidelity. Residuals less than “10⁻⁴” opens room for ambiguity.
 - iv. Could authors discuss the basis for their choice of mesh element? How are hexahedra compared to polyhedra for CFPD simulations of small particles in terms of accuracy on particle tracing?
 - v. Table 2 in the Appendix contain a column for “Faces per cell” which is not clear or discussed anywhere in the text.
 - g. 3D reconstructed geometries from XCT image public databases: Can the range of voxel resolution used to create the XCT images be also reported if accessible?
 - h. The drag model used in the CFPD model is not discussed: How is the drag coefficient defined considering the range of Reynolds number of the fluid that are found in the HRT segments?
2. The authors presented results for submicrometer-to-micrometer size of particles:
 - a. Authors need to adjust the length scale, including in the figures, to represent a particle of 0- μm .
 - b. The models were simulated as a two-way coupling for 100,000 particle density number and an average of 0.42- μm size particle at a 90 L/min flowrate to add robustness to the models, according to L327-328. However, the ratio of particle mass to fluid mass inside the HRT computational domain is not reported. This is needed to aid establishing whether this is a dilute system (i.e., one-way coupling). And, if indeed, the system should be represented as a two-way:
 - i. What are the contributions of particles on the dynamics of the fluid? And,
 - ii. How, by quantification, is particle deposition impacted by the choice of coupling approach?

iii. How the results of this paper are compared to the findings of "previous investigations", which should also be cited in L327?

3. Lastly, despite several forces were defined to influence the particle dynamics in this work, the specific contribution of the mechanisms of deposition or transport on the total deposition of the particles is not discussed. For instance, what are the contribution of Brownian forces on particle fate versus that of inertial forces or eddies?

Reviewer #3

(Remarks to the Author)

There are a lot of positives to this manuscript. It's nicely written. After careful review, I have the following concerns about the manuscript:

1. The study's novelty may be questioned due to the existence of numerous prior works in related areas. I am not sure how this study will improve the knowledge of the field.
2. Reliance on generic mathematical models may not fully capture individual physiological variances.
3. Though large, the sample size does not represent global population diversity.
4. The practical impact on current clinical practices is not fully demonstrated.
5. Computational models require thorough validation, with a more detailed discussion of potential errors.
6. The study could be more explicit about the next steps and potential improvements, with clear recommendations for further research.

Version 1:

Reviewer comments:

Reviewer #1

(Remarks to the Author)

Thank the authors for their responses to my previous comments and revising their manuscript. All my questions and concerns have been properly addressed in the updated manuscript. The revised manuscript is significantly improved in quality, therefore the publication of this manuscript on Communications Engineering is recommended.

Reviewer #2

(Remarks to the Author)

Remarks to the author(s):

Thank you for your comments and addressing some of my observations.

Regarding (1.a): There are still some confusing narrative between CFD and CFPD. CFPD treats the particle as a discrete phase so it is purely Lagrangian based (in an Eulerian mesh). CFD when directed to particle studies treats this species as a gas phase, using the Eulerian method (i.e., the discrete method is not implemented), lines 61-64 showed some ambiguity in that regards. Lines 64 to 70 seemed unnecessary as it should be focused on the approach that better describes the dynamics between the fluid and the particles for the studied cases scenarios.

Lastly, regarding (3), possible deposition mechanisms are not discussed in the latest manuscript. For instance, if particles can indeed deposit by Brownian diffusion, the particle drag model should account for that, and Stokes-Cunningham drag model becomes relevant. If some assumptions were made in the models, it should clearly be stated in the manuscript.

Version 2:

Reviewer comments:

Reviewer #2

(Remarks to the Author)

Point-by-point response to the referees' comment on manuscript COMMS-ENG-23-0605-T.

The point-by-point response to the referees' comments is organized in the following format:

- Black text: Original comments and questions from the reviewers.
- Blue text: Our responses to the referees.
- Red text: Original manuscript text.
- **Highlighted red text**: Added text to the resubmitted manuscript.
- ~~Strike-through red text~~: Deleted text from the original manuscript.

There were some minor formatting changes throughout the original manuscript and the supplemental text to comply with the Nature Communications Engineering Journal requirements that were not highlighted during the review process.

1 Reviewer #1

In this paper, the authors introduce a comprehensive computational framework to generate patient-specific particle deposition profiles and radiation dose maps from CT scans, using computational fluid-particle dynamics (CFPD) simulations and Monte Carlo radiation transport. The study analyzed a large database of 3D human respiratory tract (HRT) geometries reconstructed from CT scans, clustered them based on relevant parameters using machine learning, and performed CFPD simulations on representative geometries. The paper quantifies inter-subject variability in particle deposition patterns and airflow structures due to differences in HRT geometry. It also couples the CFPD particle deposition profiles with the PHITS Monte Carlo code to generate detailed radiation dose maps, highlighting non-uniform energy deposition, and demonstrates the importance of accounting for anatomical variations in HRT position and radiation dosimetry, going beyond the generic ICRP models. The claims in this paper are largely novel. While previous studies have explored CFPD simulations and CT-based HRT geometry reconstructions, this is one of the first large-scale studies accounting for realistic population diversity and integrating it with radiation dosimetry. The coupling of CFPD with Monte Carlo for subject-specific dosimetry, automated geometry reconstruction, and machine learning-based clustering are all novel aspects. The paper is of significant interest to researchers working on respiratory dosimetry, aerosol deposition, radiation protection, and related fields. The computational framework and insights into inter-subject variability can contribute to more accurate risk assessment and personalized therapeutic approaches. The paper has the potential to influence thinking in the field by highlighting the limitations of generic ICRP models and the need for subject-specific dosimetry approaches, especially for scenarios involving diverse populations exposed to radioactive aerosols. Based on the novelty, significance, and convincingness of the claims, as well as the potential impact on the field, I would recommend that the authors be encouraged to consider a revision for addressing the potential concerns listed below. With appropriate revisions, this study could make a valuable contribution to the field of respiratory dosimetry and radiation protection.

We sincerely thank Reviewer #1 for their thorough evaluation of our manuscript and for recognizing our work's novelty and potential impact on respiratory dosimetry, aerosol deposition, and radiation protection. We appreciate the detailed questions and concerns, which were invaluable in refining our study.

We have carefully considered each point raised and made comprehensive revisions to address the concerns and suggestions. Below, we provide a point-by-point response to each comment, detailing the specific changes made to the manuscript and how they enhance the clarity, rigor, and overall contribution of our work.

1. For the CT scan dataset, "542 reconstructed 3D geometries of the lower human respiratory tract (HRT) from subjects aged 2 to 90 years, with 54% males and 46% females, including 88 pediatric subjects", among all the subjects, are their lungs all in healthy condition or some are diseased lungs? Would there be any other conditions that may affect the morphology of the lower HRT?

The CT scans were acquired from different CT scan datasets with different conditions; some were healthy subjects, while others were affected by some diseases. Some of the patients' diseases affected the morphology of the lower and upper HRT, but that is a feature we wanted to include in the work since this research

accounts for a broad-range population, with a disease not being an excluding condition for the analysis. We acknowledge that if this work were to be tailored only for frontline responders in the case of an emergency, only healthy subjects must be included in the analysis. Below, we provide a detailed description of each dataset and how the manuscript was modified according to that comment:

Exact09 database: "[...]The conditions of the scanned subjects varied widely, ranging from healthy volunteers to patients showing severe abnormalities in the airways or lung parenchyma[...]"

Pediatric-CT-SEG database: "[...]The datasets represent random pediatric cases from Children's Wisconsin based on routine clinical indications.[...]No medical or diagnostic data are available for any patient dataset.[...]"

Lung-PET-CT-Dx database: "[...]This dataset consists of CT and PET-CT DICOM images of lung cancer subjects. [...]Patients with Names/IDs containing the letter 'A' were diagnosed with Adenocarcinoma, 'B' with Small Cell Carcinoma, 'E' with Large Cell Carcinoma, and 'G' with Squamous Cell Carcinoma.[...]"

COVID-19-AR database: "[...]All data were collected from hospitalized patients with a positive COVID-19 lab test (PCR) verified diagnosis who had imaging studies within eight days prior to diagnosis and at least one imaging study post-diagnosis.[...]"

CT vs PET Ventilation Imaging database: The health information of the 16 patients is given in the form of a table. The 16 male and female patients included in this database all had mild, moderate, or severe chronic obstructive pulmonary disease and impairment of the lungs' carbon monoxide diffusing capacity.

According to that comment we modified L362 to L370 of the original manuscript:

- From the EXACT'09 challenge, 46 de-identified chest CT scans were obtained (From the trachea up to the 5th to 7th generation of bronchi, depending on the quality of the CT scan). The conditions of the scanned subjects varied widely, from healthy volunteers to patients with severe abnormalities in the airways or lung parenchyma.
- Pediatric Chest/Abdomen/Pelvic CT Exams with Expert Organ Contours (Pediatric-CT-SEG) from The Cancer Imaging Archive (TCIA) represent 359 random pediatric cases based upon routine clinical indications. No medical or diagnostic data are available for any patient dataset.
- A Large-Scale CT and PET/CT Dataset for Lung Cancer Diagnosis (Lung-PET-CT-Dx) from TCIA. All the 355 patients in the database were diagnosed with Adenocarcinoma, Small Cell Carcinoma, Large Cell Carcinoma, and Squamous Cell Carcinoma.
- Chest Imaging with Clinical and Genomic Correlates Representing a Rural COVID-19 Positive Population (COVID-19-AR) from TCIA includes data collected from 105 hospitalized patients with a positive COVID-19 lab test verified diagnosis, with imaging studies conducted within eight days prior to diagnosis and at least one imaging study post-diagnosis.
- CT Ventilation as a Functional Imaging Modality for Lung Cancer Radiotherapy (CT vs PET Ventilation Imaging) from TCIA includes health information of 16 patients, all of whom had mild, moderate, or severe chronic obstructive pulmonary disease and impairment in the diffusing capacity of the lungs for carbon monoxide.

2. On page 3, it states that "Only four complete HRTs were connected using Blender", can the authors provide more details on criteria used for selecting those four HRTs? What are the upper HRTs' parameters that matter most regarding the particle deposition?

This is an excellent question that we did not address in detail in the original manuscript due to the complexity of the answer and the overall length constraints. The reconstructed HRT geometries for the upper and lower respiratory tracts were derived from anonymized datasets, limiting the available patient information. Ideally, we would have matched a single patient's head-neck CT scan with their corresponding chest CT scan, but this was not feasible due to the separate databases for each CT scan region of interest.

Therefore, our criteria were as follows:

1. We initially filtered by sex and similar weight (± 10 kg). Once a match was identified, we individually examined the 3D reconstructions of both the lower and upper HRT and selected the most similar in tracheal diameter. We did not use age for matching since it was a restrictive criteria given that we had only 40 3D reconstructions of the HRT.

2. For potential candidate pairs, we referred back to the source CT scans to ensure that the chest CT scan region overlapped with the head-neck CT scan region. If no overlap was present, the pair was discarded to avoid missing segments in the full HRT reconstruction.
3. Using coronal cuts and anatomical reference points, we matched the upper and lower respiratory tracts from the CT scans. This step was crucial for accurately aligning the 3D reconstructions and ensuring proper integration by removing overlapping sections.

The critical parameters for the upper HRT concerning particle deposition include (H. Calmet et al., 2018; Y. Feng, et al., 2018):

- The shape and dimensions of the nasal passages, being the most critical the nasal valve area, turbinate structure, and nasal septum deviation, significantly impact airflow patterns and particle trajectories.
- The mode (nasal vs. oral) and pattern of breathing (e.g., tidal volume, inhalation rate) alter deposition distribution, with nasal breathing typically resulting in higher deposition in the upper airways.

The manuscript was modified in L112 and L113 in the original document as follows:

Only four complete HRTs were connected using Blender, since it was a time consuming task, the criteria for connecting the upper and lower HRT it is explained in detail in Appendix II, Geometry Pre-Processing section.

The Supplemental Material was modified after L125 in the original document as follows:

The process of matching the upper and lower HRT was challenging due to the 3D geometrical reconstruction of those parts coming from different subjects. Therefore, the criteria to reconstruct the four geometries in this work were as follows:

1. Initially, it was filtered by sex and similar weight (± 10 kg). Once a match was identified, we individually examined the 3D reconstructions of the lower and upper HRT and selected the most similar tracheal diameter. We did not use age for matching since it was a restrictive criterion given that we had only 40 3D reconstructions of the HRT.
2. For potential candidate pairs, we referred back to the source CT scans to ensure that the chest CT scan region overlapped with the head-neck CT scan region. If no overlap was present, the pair was discarded to avoid missing segments in the full HRT reconstruction.
3. Using coronal cuts and anatomical reference points, we matched the upper and lower respiratory tracts from the CT scans. This step was crucial for accurately aligning the 3D reconstructions and ensuring proper integration by removing overlapping sections.

3. On page 5 line 131-141, there seems 2 redundant groups, specifically “mean t_d and mean b_a (mm)”.

Thank you for catching this mistake. We deleted the redundant groups. We have also added a table detailing the average sizes for each group to make it more clear.

Based on this comment, we deleted L136 and L137 from the original manuscript and added a table:

- mean t_d and big b_a (mb): This cluster comprises subjects with an average tracheal diameter and a large bronchial angle.
- mean t_d and mean b_a (mm): Subjects in this cluster have average values for both tracheal diameter and bronchial angle
- mean t_d and mean b_a (mm): Subjects in this cluster have average values for both tracheal diameter and bronchial angle.
- mean t_d and mean b_a (mm): Subjects in this cluster have average values for both tracheal diameter and bronchial angle.

Table 1: Range of carina angle and trachea diameter for each group in the k-mean clustering. The mean value and the standard deviation are displayed in the table, the values in parenthesis are the minimum and maximum value within the cluster.

Group	Carina Angle [degrees]	Mean Trachea Diameter [mm]
bs	76.81±2.42 (81.2-71.4)	17±0.5 (17.7-16.2)
bm	80.9±3.57 (87-76.4)	17.3±0.6 (18.2-16.1)
bb	97.83±3.53 (104-91.6)	18.2±0.3 (18.6-17.8)
mb	95.18±3.15 (101-90.6)	17±0.4 (17.6-16)
mm	79.1±3.5 (84.5-73.6)	16.2±0.4 (16.9-15.9)
ms	69.23±3 (73.2-64)	14.9±0.4 (15.9-14.3)
sb	88.57±3.6 (93.2-81.7)	15±0.5 (15.8-14)
sm	82.65±3.36 (89.7-78.4)	14.3±0.5 (15.5-13.7)
ss	67.63±3.2 (73.1-61.4)	13.46±0.1 (13.7-13.4)

4. Can the authors provide the definition of base size used in mesh independence test, and details on how it was used in the mesh generation?

The concept of base size is the same for both CFD/CFPD solvers (StarCCM+ and OpenFOAM), but the base element and mesh generation process differs in each case. Below, we provide an explanation of each concept in the mesh generation process and how it was used for the mesh independence test.

StarCCM+: In StarCCM+, the base size is a parameter that dictates the overall size of mesh elements. It serves as a reference length scale from which other mesh element sizes are derived. The base size controls the element density and distribution across the computational domain.

Base Size Definition for StarCCM+: The base size in StarCCM+ is set as a global parameter that determines the initial tetrahedral cell size before any refinement to a polyhedral mesh or local sizing controls are applied. For further calculations, the base size was used as a characteristic length for our mesh independence test. The grid independence studies found that 1.5 mm is enough to achieve a converged simulation. To ensure safety, we set the base size to 1mm for the simulations presented with StarCCM+ in this work.

Polyhedral Meshing: In StarCCM+, polyhedral meshing was utilized due to its superior accuracy and convergence properties compared to tetrahedral or hexahedral meshes. The polyhedral mesh starts with a polyhedral cell structure derived from a tetrahedral base. The base size controls the initial tetrahedral mesh before converting to polyhedral cells.

Mesh Refinement: The maximum tetrahedral size was set to 100% of the relative base size, meaning that the largest allowable cell size in the mesh generation process is equal to the base size. This approach ensures that the initial cell size is uniformly distributed throughout the domain, providing a consistent resolution before applying additional local refinements. Typically, the maximum tetrahedral size is larger than the base size. However, this will include an extra parameter that is difficult to quantify in the convergence studies and is usually not explicitly explained in other similar research articles.

Local Refinements: Specific regions of interest, such as boundary layers or areas with high gradients, were further refined using relative sizing controls, thin layer meshing, and prism layers. The local refinement allows for finer resolution in critical regions while maintaining a manageable overall cell count.

OpenFOAM (SnappyHexMesh): In OpenFOAM, the SnappyHexMesh tool is used for mesh generation, which operates differently from StarCCM+ but similarly relies on a base size parameter for initial mesh control.

Base Size: In snappyHexMesh, the base size is defined by creating a background hexahedral mesh using the blockMesh tool. This tool sets the initial size of the hexahedral cells before any refinement or snapping processes. Here, there is a distinction between the automated meshing tool we created and presented in the paper and the mesh independence test. When we refer to base size, it is the base size of the cubic-like hexahedron element. This base size was defined manually for the meshes used for the mesh independence test, as explained in L150 to L152 of the original manuscript. This is further referred to with more detail from L20 of the original supplemental material. Regarding the automatized tool we developed to generate the meshes, we take the hexahedron base size to be the size of the smallest outlet (smallest bounding cube). The size of the smallest outlet in the 3D models we have is always

smaller than 1.5mm, but for the sake of safety, we added an if statement. In pseudocode, looks like `if(min(outletSizeList)>1.5: baseSize = 1.5; else: baseSize = min(outletSizeList)`.

The application in Mesh Generation is as follows:

Hexahedral Meshing: SnappyHexMesh begins with a background hexahedral mesh, where the base size determines the initial cell size. This uniform grid is the starting point for subsequent mesh refinement and snapping processes. Feature Refinement: The snappyHexMesh process includes several refinement steps, including castellated mesh generation, surface snapping, and layer addition. Each step refines the mesh based on the initial base size and predefined refinement levels. Local refinements are applied to capture geometric features, boundary layers, and regions with expected high gradients. Local Refinements: The surfaceFeaturesExtract tool guided the implementation of additional local refinements in areas of high-flow gradients or complex geometry. This step ensures the mesh resolution is adequate for accurately capturing flow details while maintaining computational efficiency.

We conducted a series of simulations with varying base sizes for the mesh independence test to determine the optimal mesh resolution. The test involved refining the base size and observing the impact on crucial flow parameters, ensuring that further refinements did not significantly alter the results, achieving mesh independence.

According to these comments, we modified the original manuscript in L150.

[...]A high-fidelity simulation with a 0.054 mm mesh base size was performed to set a benchmark for convergence. In OpenFOAM, the element base size (i.e., mesh size) was defined by creating a background hexahedral mesh using the blockMesh tool. This tool sets the initial size of the hexahedral cells before any refinement or snapping processes. Similarly, the element base size in StarCCM+ was set as a global parameter that determines the initial tetrahedral cell size before any refinement to a polyhedral mesh or local sizing controls was applied. Convergence metrics—Order of Grid Convergence (calculated via a least-squares fit), Grid Convergence Index (GCI), and Asymptotic range of convergence—were used to assess the simulations [...]

We also further modified the supplemental material in L21 and L134 from the original document as follows:

[...]Element base sizes spanned a range from 1.84 mm for the coarsest mesh to 0.087 mm for the most refined mesh. In snappyHexMesh, the base size is defined through creating a background hexahedral mesh, using the blockMesh tool. Five intermediate meshes were created to augment our dataset, resulting in an ensemble of seven distinct computational domains. [...]

[...] The size of the cubic element is determined by the size of the minor outlet, ensuring that at least one element intersects each surface, as recommended by the SnappyHexMesh user manual. If the size of the minor outlet exceeds 1.5 mm, we set the cubic element to be 1.5 mm, based on our mesh independence studies. In StarCCM+, the element base size is a global parameter that sets the initial tetrahedral cell size before any refinement to a polyhedral mesh and local sizing controls are applied. [...]

5. The author explained explicitly the importance of choosing appropriate wall functions when using SST LM RANS model. Can they justify their reason for applying “low Reynolds wall function for k and a regular wall function for ω was used at the walls” in their modeling work?

In OpenFOAM, careful consideration must be made about wall functions, considering the y^+ . In the case of using StarCCM+, the software has a feature called “All y^+ treatment,” which will select the appropriate wall treatment. To answer the question from Reviewer #1, we chose such wall functions because we use a fine mesh (relative to the turbulent zones), where the y^+ is well below 1 in most cells and close to 1 in some cells. Therefore, for this RANS turbulence model, the most appropriate and stable wall functions in OpenFOAM are `kLowReWallFunction` and `omegaWallFunction`. It is worth noting that the `kLowReWallFunction` boundary condition provides a wall constraint on the turbulent kinetic energy for low- and high-Reynolds number turbulence models from the user manual. To avoid further confusion, we modified the L143 of the supplemental material.

[...]The outlets’ k and ω condition is the Neumann boundary condition equal to zero. While in StarCCM+, the wall treatment is done by activating the “All y^+ treatment” feature, in OpenFOAM, this requires manual handling. For the turbulent kinetic energy k , the `kLowReWallFunction` was used, which can handle both low and high-Reynolds regimes. For ω instead, the standard `omegaWallFunction` was used. This combination of wall functions is the most stable for the family of SST transitional models. A low Reynolds wall function for k and a regular wall function for ω was used at the walls.

To avoid any confusion, the `kLowReWallFunction` boundary condition provides a wall constraint on the turbulent kinetic energy for low- and high-Reynolds number turbulence models, as per the user manual. Additionally, using a low Re model for k is critical to the model due to the intermittency parameter. The fluid in our study transitions between turbulent and laminar regimes, and a model that accounts for this transition without numerical instability is needed. Near the walls, at low Reynolds numbers, the turbulence kinetic energy can exhibit problematic behavior if not adequately modeled, leading to inaccuracies or solver failures. [...]

6. Have the authors considered validation of particle deposition profiles and radiation dose maps against experimental data or clinical measurements for selected HRT geometries? For Figure 5, were the clinical data based on in vitro test or in silico results? Was the same lung geometry used in previous studies? If different lung model was used, how to justify the effect of HRT morphology on particle deposition pattern?

We have considered validation against both computational and experimental work regarding the particle deposition profiles. From the studies listed and referenced in Figure 5, Zhou et al. (2011) used three different models and did both in-vitro (i.e., label with Exp) and in-silico (i.e., label with Num) experiments. The three models varied in complexity: USPIP, a simplified idealized mouth-throat model (L-shaped tube); UofA, a simplified mouth-throat HRT; and LRRI, a realistic cast of the mouth-throat model. Arsalanloo et al. (2022) performed CFPD in a realistic lower HRT but with a limited number of bronchi generations. Bowes S. M. et al. (1989) did experimental work, measuring particle deposition in the mouth tract of human subjects, while Cheng, Y-S. et al. (1999) performed in-vitro experiments in a realistic mouth+lower HRT cast up to the 4th generation of bronchi. In the same line, Kim, Y.H. et al. (2022) conducted CFPD experiments in a realistic mouth-trachea and lower HRT model for power inhalers and reported their experiments using the impaction parameter as a metric. Lastly, the results from Longest, P. W. et al. (2008), Matida E. et al. (2004), and Zhang, Z. et al. (2004) come from CFPD studies using an idealized human mouth-throat model.

We are unsure if the reviewer #1 with "same lung geometry" refers to one of the geometries used in the studies presented in Figure 5 from other authors or if it is about one of the geometries in our study. We will answer both, hoping one of those address the question. Regarding the lung geometries used in previous studies, mentioned in Figure 5, an idealized human respiratory mouth-tract model has been used across multiple authors. In two works, it was also attached to the idealized human respiratory lower tract from ICRP Publication 66 to represent the human HRT further. Regarding the geometry used in the presented work, it was unique, and for each geometry, to the best of our knowledge, there is no other study using the same databases to reconstruct 3D HRTs. Since a different model was used from what previous studies have reported, we also opted to use the impaction parameter (IP) to quantify the deposition efficiency. Using the IP as a metric, the particle deposition efficiency can be compared across different particle sizes and breathing conditions. Leaving apart the numerical and experimental errors that could be introduced, the differences between authors are due to the different geometries used. The takeaway of the validation is that, in general, particles with a more significant impaction parameter should have higher deposition efficiency due to inertial forces being the primary mechanism for deposition.

To further address any validation question, we extended the validation work to compare the most similar realistic HRT geometry among the research done in CFPD in full HRT geometries. We included the details of this validation in the modified manuscript and at the end of our answer to this question.

Lastly, comparing or validating radiation dose maps is an ongoing work that is challenging due to the novelty of this type of work. Only one previous study tried to address a similar problem using idealized geometry and a limited-resolution voxel phantom. The correct way of addressing this would be to do a PET-CT scan with the appropriate radioactive aerosols in a physical phantom that has the airways included, but due to the complexity of such an experiment, it is out of the scope of this work.

According to this comment, we modified L188 to L192 of the original manuscript:

~~First, verification~~ To proceed with the verification and validation of the method, the deposition efficiency was compared against previous researchers' data, using the deposition efficiency against the impaction parameter. The impaction parameter is defined as $d_p^2 Q$, where d_p is the particle diameter and Q is the airflow used. The rationale behind using the impaction parameter as a metric was that the particle deposition efficiency could be compared across different studies independently of the particle size and breathing conditions used. Ignoring potential numerical and experimental errors, the difference between studies was due to the different HRT geometries employed.

The studies listed in Figure 5, Zhou et al. (2011) used three different models and did both in-vitro (i.e.

label with Exp) and in-silico (i.e. label with Num) experiments. The three models varied in complexity: USPIP, an idealized mouth-throat model (L-shaped tube); UofA stands for a simplified mouth-throat HRT, and LRRI a realistic cast of the mouth-throat model. Arsalanloo et al. (2022), performed CFPD in a realistic lower HRT but with a limited number of bronchii generations. Bowes S. M. et al. (1989) did experimental work, measuring particle deposition in the mouth tract of human subjects, while Cheng, Y-S. et al. (1999) performed in-vitro experiments in a realistic mouth+lower HRT cast, up to the 4th generation of bronchii. Along the same line, Kim, Y.H. et al. (2022) conducted CFPD experiments in a realistic mouth-trachea plus lower HRT mode. Lastly, the results from Longest, P. W. et al. (2008), Matida E. et al. (2004), and Zhang, Z. et al. (2004) come CFPD from studies using an idealized human mouth-throat model, while Liu et al. (2023), used a combined mouth-throat and lower HRT model.

Regarding the airflow value used in this study for the verification, it was chosen to be the mean airflow in the respiratory cycle. [...]

A comparison with Dong, J. et al. (2019) was conducted to validate the CFPD methodology in this work further. The study by Dong et al. was an appropriate comparative validation, having employed a very similar 3D HRT model to one of the four full HRT geometries reconstructed in this work, in addition to a similar turbulence model and boundary conditions. The simulation parameters adopted as listed in Figure 1, to exactly match the conditions in the CFPD experiment from Dong, J. et al. (2019), including the particle physics and coupling method. The results from the validation can be seen in Figure 1. The conditions were set to match Dong's, J. et al. (2019) work, where they used a constant inflow rate of 50 L/min with a 1 s injection and 2 s breathing cycle. Simulation parameters were adapted to match those used in Dong, J. et al. (2019), including a particle density of 1,000 kg/m³, one-way coupled particle-turbulence interactions, and the inclusion of drag and Brownian forces. The experimental setup and turbulence model ($k-\omega$ SST) were also replicated to ensure accurate comparison.

Figure 1: Comparison of particle deposition fractions between this research CFPD model and the experimental results of Dong, J. et al. (2019). The left graph shows the deposition fractions for the lower HRT, while the right graph shows the deposition fractions for the upper HRT.

7. In Figure 5 right plot, it seems that deposition efficiency in sm is hardly affected by the impaction parameter within 10 to 1e10? Do the authors have any possible explanation regarding this finding?

First, we would like to clarify that there was a typo in the labels of that graph. The graph on the left has the correct labels, but the graph on the right should have shown "sb, ms, bm, bb" instead of "mb, mm, bs, sm," respectively. We did not switch the index in the code for the labels, but the data displayed confirmed that it was accurate. Now, the manuscript displays the correct labels corresponding to the HRT geometries described in the work. Regarding the question, we did not want to jump to any conclusion based on one single case, but that geometry had the most negligible secondary flow produced in the trachea and the main bifurcation (From Figure 4 in the manuscript and Figure 3 in the supplemental). The secondary flow is why the largest particles are deposited due to inertial effects. Since this geometry has a limited secondary flow and no large eddies are generated, we attribute that effect to the more uniform particle deposition seen in Figure 5, enhancing other deposition mechanisms. Nevertheless, further analysis would be required to demonstrate that behavior. Overall, we find the deposition efficiency to follow the same patterns as other researchers' results when compared using the impaction parameter, which was the purpose of that validation.

8. The paper does an adequate job of discussing the claims in the context of previous literature, citing relevant studies on CFPD simulations, CT-based geometry reconstructions, and radiation dosimetry models.

However, a more comprehensive discussion of previous subject-specific dosimetry efforts and their limitations could further contextualize the novelty of this work.

While we acknowledge that the discussion on previous subject-specific dosimetry work seems narrow, the authors conducted an extensive review of the existing literature before submitting this manuscript, and we found few comparable studies. Specifically, we have cited a significant prior study that is vaguely similar to our work and is detailed in lines L50 to L57 of the manuscript. In those lines, we discuss the limitations of a previous attempt at more precise dosimetry models from inhaled aerosols and how our approach addresses these gaps. Our work builds upon these foundational studies by integrating advanced subject-specific CFPD simulations using high-resolution CT-based geometry reconstructions, offering a more precise and individualized assessment of aerosol deposition and radiation dose distribution.

2 Reviewer #2

Advanced Patient-Specific Computational Insights into Aerosol Deposition and Radiation Exposure in Diverse Human Airways.

This work highlights the importance of studying the contribution of geometrical characteristics of human respiratory tracts (HRT) that are specific to individuals on subject-oriented radiation dosimetry tools. These geometrical variances can affect the distribution of deposited radioactive aerosol upon radiological exposure on HRT walls, which is the focus of the work. In the paper, a synergistic computational approach that uses (1) the numerical methods of computational fluid dynamics (CFD) and computational fluid and particle dynamics (CFPD) on reconstructed 3D HRTs are implemented to simulate the deposition of inhaled aerosols on HRT walls. This approach is coupled with (2) Monte-Carlo simulations to generate energy deposition maps; and (3) the random forest regression learning algorithm to predict individual weight and age from reconstructed HRT geometries, identifying the most significant features in the HRT different geometries. In this work, findings support the significance of personalized dosimetry studies due to particle deposition being highly dependent on small variations of the HRT geometry. The methodology approach presented is found to be of high interest in the breath aerosol research community in general and, more specifically, drug delivery in addition to the current application. Although, the paper is relatively well organized; the level of a comprehensive computational setup and analysis is substantially limited with some incongruencies as well as inadequate in details, consistency, and clarity; particularly in the field of (1) above. Observations are in detail below and based on those, this paper is not recommended for publication in this journal under the present version.

We sincerely thank Reviewer #2 for their thorough and detailed review of our manuscript and overall comments. Reviewer #2 insightful suggestions have been of great value in identifying areas for improvement, particularly for the CFD and CFPD parts. We appreciate Reviewer #2 time and effort invested in providing such comprehensive feedback. While writing the initial manuscript, we addressed multiple concerns that reviewer #2 brought up in the review process. However, we needed to trim the text later due to the journal's word limit.

We acknowledge that the manuscript in its previous version may have needed to meet the high standards required for publication in this journal under Reviewer #2 criteria. However, we have reviewed your feedback thoroughly and made substantial revisions to address the issues raised. We are resubmitting the manuscript as recommended by the Editor-in-Chief. We hope the reviewed version of the manuscript now meets the journal's high-quality standards and aligns with Reviewer #2 criteria.

Our revisions focused on clarifying the computational methodologies and improving the organization and detail of the model inputs. We have provided detailed explanations of our numerical methods, including the distinctions between CFD and CFPD, the implications of using RANS versus LES models, and the specifics of our model inputs and boundary conditions. We have included additional figures and tables to support our assumptions and ensure reproducibility. Below, we provide a point-by-point response to each of Reviewer #2 comments, detailing the specific changes made to the manuscript:

1. This paper is presented in such a fashion that even when the journal is oriented to the engineering community, the reader must be quite familiar with the specific numerical methods used and infer important model information that are not found in either the paper or its supplementary.

We understand that the paper is oriented in a fashion that the reader must be familiar with the specific

turbulent and particle coupling models. We tried in the original manuscript to explain each model parameter in detail to ensure reproducibility. However, based on the reviewer #2 comments below, we acknowledge that some details may have been omitted, making it challenging to ensure reproducibility. We hope the new version of the manuscript clarifies the models used and provides all the necessary details to ensure reproducibility across the community. On top of that, all the models used for StarCCM+ and OpenFOAM are available at request for anyone trying to reproduce this work. The automated CFD/CFPD workflow makes this work reproducible and will provide all the necessary setups for both OpenFOAM and the newer versions of StarCCM+. By providing the geometry, the user can automatically set up a converged simulation by running a Python script for OpenFOAM (tested latest in OpenFOAM v2112 version) and StarCCM+ (last tested in StarCCM+ v2310, v2306, v2302). That makes this work oriented to both general CFD/CFPD users, with limited knowledge of either tool or more advanced users who would like to play setting their parameters.

a. The numerical methodologies of CFD and CFPD are interchanged back and forth without making any distinction between the two of those. CFD is not defined in L41 and the primary differences relevant to the field of study between these two methodologies are missing. Authors should define both methodologies and state what the advantages of CFPD are in terms of predicting particle trajectory and distribution and what the advantages of CFD are in terms of fluid dynamics and how they can be coupled. We apologize for the oversight in not defining CFD in L41. Based on your comments, we modified the original manuscript starting on L36 as follows; please note that [citation] means that there is a citation in the reviewed manuscript that we did not want to include here to not add a bibliography section and improve readability.

[...] Computational Fluid Dynamics (CFD) is the foundational approach for modeling airflow in the HRT proposed in this study. By exploiting the Navier-Stokes equations, different models investigate different effects on human airways. Specifically, CFD has the advantage of visualizing local phenomena that cannot be seen otherwise by conducting *in-vitro* or *in-vivo* experiments. A plurality of studies employing CFD have ranged from analyzing intricacies of the nasal airflow under different breathing conditions using realistic nasal models [citation] assessing the airflow in subjects with rhinosinusitis [citation], to investigating how the laryngeal jet generated in the trachea affects the airflow downstream in the intrathoracic human airways [citation]. When particles are introduced into the system, the Navier-Stokes equation for the flow field needs to be coupled with the equations of motion for the particles. To effectively model aerosol transport in the human airways, consideration of the aerodynamic size of particulate matter is required to determine the interaction mechanism. Particles ranging from 10 nm to 100 μm may be represented as solid or droplet phase geometries and may change size due to evaporation or condensation. The high humidity in the respiratory tract can cause hygroscopic particles to grow. Particles undergo various physical interactions with the fluid, leading to different deposition mechanisms along airway surfaces, including impaction, sedimentation, and diffusion. The impaction mechanism affects particles with sufficient momentum, causing deposition at airway branches due to abrupt directional changes in airflow. Sedimentation occurs when particles with enough mass deposit due to gravity after traveling in the airways. The Brownian motion mechanism causes deposition by random diffusion, particularly for nano-sized particles.

In transitioning from CFD to CFPD, coupling the Navier-Stokes equations with the equations of motion for particles is critical for the simulation of fluid-particle interactions, accounting for the various mechanisms of particle deposition influenced by airflow and different particle sizes. When coupling the flow field with particles, there are two primary approaches: the Lagrangian approach, which treats particles as discrete points moving based on interactions within the flow field and other particles, and the Eulerian model, which assumes particles are a continuous medium with representative properties. In both cases, the flow field is resolved on an Eulerian mesh [citation]. The one-way coupled Euler-Lagrange method assumes particles do not affect the carrier flow, meaning particles move without providing any feedback to the airflow. The two-way coupled Euler-Lagrange method considers the impact of particles on the fluid motion, requiring a detailed coupling of particle forces into the airflow momentum equations. For more complex interactions, such as particle-particle collisions, the four-way coupled Euler-Lagrange method adds terms to account for these interactions [citation]. An alternative Euler-Euler approach, suitable for spherical and quasi-spherical particles smaller than 100 nm, uses mass transfer equations to model particle behavior, assuming isotropic diffusion and turbulent dispersion.[...]

b. Elaboration is needed when comparing RANS vs. LES models for HRT flow characterization. What are the implications of selecting one method over the other in terms of accuracy per HRT segment as well

as computational time?

In this regard, Koullapis P. G. et al. (2018) extensively analyzed and compared LES models versus RANS models. Based on this comment and Koullapis P. G. et al. (2018) work, we added an explanation after L45 from the original manuscript. Regarding the computational time of LES versus RANS, it is a difficult question to answer; from previous experience and published works in other fields, using LES takes at least an order of magnitude more computational time than a RANS simulation. Previous research that used LES and compared LES versus RANS did not report the simulation time, making a direct comparison of LES vs RANS in the HRT not traceable. In the presented research work, we used between 48 and 128 cores for the simulations, depending on the cluster used. The nodes contained Dual Xeon Platinum 8268 or Dual Intel Xeon Gold 6226 CPUs. Using these settings, CFD simulations without particles took around 8 hours to finish with the converged mesh settings, and the CFPD simulations took between 24 hrs and seven days, depending on the geometry used, the total particle numbers injected, and if using OpenFOAM (slower) or StarCCM+.

Specifically, after L45, we modified the manuscript following the reviewer’s comments and questions.

[...] In this regard, Reynolds-Averaged Navier-Stokes (RANS) has proven to be as effective as Large-Eddy Simulation (LES) models. RANS models average the effect of the turbulence, while LES models resolve large-scale turbulent structures directly. LES methodology uses filtered Navier-Stokes equations and models only the smaller scales of turbulence, hence the less energetic ones. Koullapis P. G. et al. (2018) [citation], demonstrated that certain RANS models are as effective as LES simulations for predicting turbulence effects and particle deposition patterns, but RANS slightly overpredicting particle deposition in some cases. In the Koullapis study, computational deposition measurements employed particle diameters ranging from $d_p = 0.5 \mu\text{m}$ to $10 \mu\text{m}$, particle density $\rho_p = 914 \text{ kg/m}^3$, and a constant airflow of $Q = 60 \frac{\text{L}}{\text{min}}$, $Q = 30 \frac{\text{L}}{\text{min}}$, and $Q = 15 \frac{\text{L}}{\text{min}}$. The main constraint for less is that the computational cost of LES is significantly higher than RANS models. LES requires finer meshes and smaller temporal resolution, increasing computational time by at least an order or magnitude. This increase in computational time makes LES less feasible for routine clinical applications or large-scale studies. While significant advancements have been made, the current state-of-the-art lacks the ability to conduct personalized studies.[...]

c. Model inputs, including fluid and particle material properties used, in an organized fashion could be useful not only clarity but also to allow reproducibility.

We thank Reviewer #2 for this comment; we added a table after L417 with all the information about the models, parameters, and material properties. We hope this makes the manuscript more readable and allows reproducibility.

i. In L212, the particle dynamics are set as unsteady but not details of the particle time steps are provided per particle size distribution/injections. Moreover, it is not clear what the boundary condition describing the particle-wall interaction is. It could be inferred they are “trapped” based on L208 but this is unclear whether particles are trapped upon a single contact with the wall?

In the Table presented below and added after L417, in the CFPD Modelling section of the manuscript, we added the information about the time integration scheme used for each solver. Moreover, information about the time steps used is also provided. The time steps differed depending on whether to resolve only the lower HRT or the full HRT. In the case of the lower HRT, a time step advancement of 0.025 s. was used; instead, for full HRT, we opted for a time step of 0.015 s. to obtain a more stable solution with lower residuals per time step.

Regarding the boundary condition for the particle-wall interaction, the only options available in the solvers are “Stick” or “Rebound.” Therefore, the “Stick” option was preferred, as the particle will stick to the wall upon contact.

Thanks to comments c. and c.i., Table 2 was added to the manuscript after L417.

A summary of the models, parameters, and values used in this work is tabulated in Table 2.

d. In L143, the authors assumed a “low Reynolds” flow for their choice of turbulence treatment on the wall, which also contradicts their argument to implement a two-way coupling approach. What would the basis be for such assumptions having a peak of 90 L/min flowrate in the domain studied? A contour-plot figure showing the fluid Re numbers throughout all HRT segments is needed to support such assumption and prevent subjectivity.

We also partially addressed this question to Reviewer #1, and we understand where the confusion came from. We acknowledge that writing “low Reynolds wall function for k ” caused the reader to think the flow

Table 2: List of models, parameters, and boundary conditions for the airflow and particles in the simulations presented in this work. A more detailed explanation on the selection of this parameters can be found in the Supplemental Material, CFPD Modelling section.

Description	Parameter Value/Condition (Flow Solver)	Description	Parameter Value/Condition (Flow Solver)
Turbulence Model	$k - \omega$ SST LM	Wall condition for \vec{u}	No slip boundary condition
Inlet Boundary conditions for \vec{u}	Variable flow rate: $Q = Q_{\text{max}} \sin(\frac{2\pi t}{T})$	Airflow density and temperature	1.177 kg/m ³ ; 300 K
Outlet Boundary Conditions \vec{u}	Zero gradient	Particle solver	MPPICFoam (OpenFOAM); Lagrangian Multi Phase (StarCCM+)
Inlet Boundary conditions for p	Zero gradient	Particle coupling	MPPICFoam (OpenFOAM); Lagrangian
Outlet Boundary conditions for p	Fixed pressure value (static pressure of the environment)	Particle distribution density function:	log-normal, $\mu = 0.42 \mu\text{m}$, and $\sigma = 3.5$ $\frac{1}{x\sigma\sqrt{2\pi}} \exp\left(-\frac{(\ln x - \mu)^2}{2\sigma^2}\right)$
Inlet Boundary conditions for k	$\frac{3}{2}(I(u_{ref})^2)$ $I = 0.04$, 4% turbulence intensity	Particle density	Iodine, $\rho = 4390 \text{ kg/m}^3$
Inlet Boundary conditions for ω	$\omega = \frac{k^{0.5}}{C_{\omega}^{0.25} L}$	Number of particles injected to the system	100,000 particles (OpenFOAM)
Inlet Boundary conditions for γ	$\gamma = 1$	Injection rate	1/(time step) per injector point (StarCCM+)
Inlet Boundary conditions for Re_{θ}	See Supplemental Material, CFPD Modelling	Particle-Wall interaction mode	Stick upon contact with the wall
Outlet Boundary conditions for $\gamma, Re_{\theta}, k, \omega$	Zero gradient	Time integration scheme	PISO algorithm (OpenFOAM); Implicit unsteady (StarCCM+)
Wall treatment for k	All y^+ treatment (StarCCM+); kLowReWallFunction (OpenFOAM)	Time step used	0.015s - 0.025 s
Wall treatment for ω	All y^+ treatment (StarCCM+); omegaWallFunction (OpenFOAM)	Total time simulated	4 s. (2 s. flow only and 2 s. with particles)

has a low Reynolds number throughout the whole domain. However, as we will explain below, that wall function works for both low- and high-Reynolds numbers.

In OpenFOAM, the wall treatment should be handled manually, and careful consideration needs to be made about the y^+ . When using StarCCM+, the software has a feature called "All y^+ treatment," which will select the appropriate wall treatment, blending different functions. The reason for choosing such wall functions is that we use a fine mesh (relative to the turbulent zones), where the y^+ is below 1 in most cells and close to 1 in some cells. Therefore, for this RANS turbulence model, the most appropriate and stable wall functions in OpenFOAM are `kLowReWallFunction` and `omegaWallFunction`.

It is worth noting that the `kLowReWallFunction` boundary condition provides a wall constraint on the turbulent kinetic energy for low- and high-Reynolds number turbulence models, as per the user manual. Additionally, using a low Re model for k is critical to the model due to the intermittency parameter. The fluid in our study transitions between turbulent and laminar regimes, and a model that accounts for this transition without numerical instability is needed. Near the walls, at low Reynolds numbers, the turbulence kinetic energy (k) can exhibit problematic behavior if not adequately modeled, leading to inaccuracies or solver failures.

To avoid further confusion, we have modified L143 of the supplemental material.

[...]The outlets' k and ω condition is the Neumann boundary condition equal to zero. While in StarCCM+, the wall treatment is done by activating the "All y^+ treatment" feature, in OpenFOAM, this requires manual handling. For the turbulent kinetic energy k , the `kLowReWallFunction` was used, which can handle both low and high-Reynolds regimes. For ω instead, the standard `omegaWallFunction` was used. This combination of wall functions is the most stable for the family of SST transitional models. A low-Reynolds wall function for k and a regular wall function for ω was used at the walls.

To avoid any confusion, the `kLowReWallFunction` boundary condition provides a wall constraint on the turbulent kinetic energy for low- and high-Reynolds number turbulence models, as per the user manual. Additionally, using a low Re model for k is critical to the model due to the intermittency parameter. The fluid in our study transitions between turbulent and laminar regimes, and a model that accounts for this transition without numerical instability is needed. Near the walls, at low Reynolds numbers, the turbulence kinetic energy can exhibit problematic behavior if not adequately modeled, leading to inaccuracies or solver failures.[...]

To further address this comment, we would like to provide a plot of the gamma function (γ parameter in the RANS $k - \omega$ SST LM model) over one selected slice to demonstrate the intermittency of this model,

this behavior is shown in Figure 2. The closer the value to zero is where the model assumes a laminar flow and the closer to one the value, is where turbulence is resolved.

Figure 2: Intermittency value for a CFPD simulation at the peak of inhalation (i.e. 90 L/min.). The closer the value to 0 is where the model assumes a laminar flow, the closer the value to 1, the flow will be treated as turbulent. For the values in between 0 and 1, the $k - \omega$ SST LM model use blend functions to treat the transitional regime.

e. L18-19 in Appendix I: There is no clarity about what the authors want to convey by “fixed pressure boundary conditions” at the outlets.

We clarified the condition by stating that the model boundary conditions were the same as the rest of the simulation. The only parameter that changed was the flow rate, which was set to a constant value of 90 L/min. We understand that stating only the pressure boundary condition can lead to confusion about the rest of the boundary conditions, so we changed L18-L19 of the manuscript as follows.

These meshes were subsequently used to solve steady-state conditions under an constant inlet airflow rate of $Q = 90$ L/min with fixed pressure boundary conditions at the outlets. The rest of the parameters for the airflow remained the same as Table 2 from the main article, with except of using SIMPLE algorithm instead of PIMPLE algorithm, since it is a steady state simulation, the specific solver used was simpleFoam in OpenFOAM.

f. “Element base size” or “mesh size” expressed as cell length is preferred as well as including the domain length, for comparison purposes, and a scale bar added to Figure 1 in Appendix 1.

Based on this comment, we changed where ”base size” was written to ”element base size” in the sentences on L150, L152, and 160 of the original manuscript. Additionally, L25 and Table 2 of the additional information were modified.

Regarding Figure 1 in Appendix 1, a scale bar was added to the figure for reference on the right side. According to this, the manuscript and supplemental information were modified as follows:

[...] An additional high-fidelity simulation was conducted using a mesh featuring an even finer element base size of 0.054 mm to establish a benchmark for convergence analysis. [...]

[...]Meshes ranged from a coarse 1.84 mm element base size to a fine 0.087 mm,[...]

[...]A high-fidelity simulation with a 0.054 mm element base size was performed to set a benchmark for convergence.[...]

i. Furthermore, in L25 in Appendix I, the basis to establish high-fidelity simulations for “base size of 0.054 mm to establish a benchmark for convergence analysis” doesn’t seem to be sufficient based merely on mesh refinement with no further information (see ii below).

Thanks for the comment, please see answer for ii. below.

ii. No element quality is reported, this is needed to validate model robustness. Also, a histogram per case studied showing the average of cell quality in the model should be included in Figure 1, Appendix 1.

We thank Reviewer #2. It was critical to have outstanding quality meshes to have successful results in the mesh independence test using seven different meshes. Moreover, all the parameters in StarCCM+ and OpenFOAM (SnappyHexMesh) were carefully tuned to obtain a mesh quality that flawlessly passed all the mesh quality checks in both tools. As Reviewer #2 recommended, we added a histogram at the bottom of Figure 1 in Appendix I. Given this modification, the manuscript was changed as follows:

L151 of the original manuscript was modified as: [...] A high-fidelity simulation with a 0.054 mm element base size was performed to set a benchmark for convergence. All the meshes were tested to pass mesh quality checks successfully, see Figure 1 in Appendix 1 for more information. [...]

The Supplemental Material L24-25 was modified to explain the mesh quality checks performed and add a histogram with the skew value of the elements in each cell as a parameter for mesh quality.

When generating the meshes, all the parameter values in SnappyHexMesh were carefully adjusted to pass quality mesh checks. All the meshes were tested under astringent quality checks that accounted for Overall domain bounding box, Mesh geometric directions, Mesh solution directions, Boundary openness, Max cell openness, Max aspect ratio, Face area magnitudes, Cell volumes, Mesh non-orthogonality, Face pyramids, Skewness, Coupled point location match, Edge length, Faces with concave angles, Face flatness, Faces with a ratio between projected and actual area ≤ 0.8 , Cell determinant (well-posedness), Concave cells (using face planes), Face interpolation weight, Face volume ratio. To report the cell quality, in Figure 3, on the bottom row, the distribution of the skew value for each cell can be seen in the form of a histogram. The skew value should be as close as possible to zero, and the average skew value is always well below the ≤ 0.3 limit.

Figure 3: A subset view of five of the seven distinct meshes employed in the mesh independence analysis is presented. The upper row depicts a broad overview of the geometric configurations, while the middle row provides a magnified view of each mesh. The distance (in millimeters) denoted corresponds to the edge length of the largest cubic hexahedron within the mesh. The two remaining mesh images are excluded for visual presentation and are consistent with the characteristics depicted in the five depicted cases. The bottom row shows the skew value for each mesh in a histogram fashion; the closer the skew to zero, the better the mesh. All the meshes scored an average skewness of less than 0.3, demonstrating the meshes' quality.

iii. L27, Appendix 1: Residual value in the CFD-CFPD models should be clearly quantified to support model fidelity. Residuals less than “10⁻⁴” opens room for ambiguity.

We thank the reviewer for this comment, but there is no ambiguity in that statement. We carefully monitored the residuals for all the simulations to ensure convergence, considering we are working with such complex geometries. Find below in Figure 4 of the residuals for the mesh with an element base size of 0.95 mm; this size was selected to plot the residuals just as a reference; all the other residuals looked very similar.

iv. Could authors discuss the basis for their choice of mesh element? How are hexahedra compared to

Figure 4: Residual values for a steady-state simulation using SIMPLE solver and an inlet flow of 90 L/min, using RANS $k - \omega$ SST LM model in a mesh with an element base size of 0.95 mm.

polyhedra for CFPD simulations of small particles in terms of accuracy on particle tracing?

Yes, we will clarify in the main article the choice of the base element for meshing purposes. We refer to a previous study by Thomas, M. L. and Longest, P. W. (2022), where they assessed the difference between polyhedral mesh and hexahedral mesh for particle deposition. The results from this study demonstrated no significant differences in the deposition efficiency between the two meshes. We modified L395 of the original manuscript by adding the following text:

[...] For the meshing process, the authors relied on hexahedral-based meshes for OpenFOAM and polyhedral meshes for StarCCM+. To perform the meshing process in OpenFOAM, the preferred tool was SnappyHexMesh, while in StarCCM+, the polyhedral volumetric mesh was chosen. The difference in choosing between a hexahedron-dominant mesh and a polyhedral mesh did not affect the particle deposition profiles, as discussed by Thomas, M. L. and Longest, P. W., where they found no significant differences in deposition efficiency values using a simplified bifurcation geometry. A more detailed explanation of the automated meshing process and calculation for the boundary conditions can be found in Appendix 2 from the Additional Information. ~~in the meshing process and boundary conditions subsection.~~ [...]

v. Table 2 in the Appendix contain a column for “Faces per cell” which is not clear or discussed anywhere in the text.

We apologize for the oversight in not discussing this anywhere else in the text. It was intended to demonstrate the quality of the hexahedral meshes of the Mesh Independence Test. One of the problems with the SnappyHexMesh meshing tool is that it does not restrict the number of faces on the element it generates to do a body-fitted mesh. Therefore, it can create elements with many faces, leading to local instabilities. By providing a good background mesh, one can avoid this problem and generate a heavy hexahedron-dominant mesh quantified by the average number of faces per cell in the mesh. On the supplemental material, L53 was modified as follows:

The faces per cell in Table 2 refer to the average number of faces per cell present in each mesh, which relates to the quality of the hexahedron-dominant mesh generated by SnappyHexMesh. The closer the value to 6, the fewer elements with more than six faces were introduced in the meshing process. Elements with multiple faces can lead to local instabilities if they are significant in the mesh, but in the worst case, 8% of the elements were not hexahedrons. In the best scenario, 99% of the elements were hexahedrons.

g. 3D reconstructed geometries from XCT image public databases: Can the range of voxel resolution

used to create the XCT images be also reported if accessible?

Since all the CT scans come from different reconstruction kernels and machine settings, we provide the voxel resolution’s mean, standard deviation, and min and max values. The values are given in Table 3 and added to L24 in the Geometry reconstruction section in Appendix II of the Supplemental Information

Table 3: Statistics for the voxel dimensions of the CT scans used in this work.

	x	y	z
Mean [mm]	0.67	0.67	0.95
Standard Dev. [mm]	0.07	0.07	0.17
Min [mm]	0.55	0.55	0.60
Max [mm]	0.78	0.78	1.25

h. The drag model used in the CFPD model is not discussed: How is the drag coefficient defined considering the range of Reynolds number of the fluid that are found in the HRT segments?

Since the particles are solid for the purpose of the work presented, we used the Schiller-Naumann Correlation to model the drag coefficient in the drag force.

The Schiller-Naumann model is appropriate for spherical solid particles. It is formulated as:

$$C_d = \begin{cases} \frac{24}{\text{Re}_p} (1 + 0.15\text{Re}_p^{0.687}) & \text{Re}_p \leq 10^3 \\ 0.44 & \text{Re}_p > 10^3 \end{cases} \quad (1)$$

where Re_p is the particle Reynolds number that is defined as:

$$\text{Re}_p \equiv \frac{\rho|\mathbf{v}_s|D_p}{\mu} \quad (2)$$

where D_p is the particle diameter and μ is the dynamic viscosity. This correlation is available only when the continuous phase is viscous. Thanks to this comment, we modified L404 of the original manuscript to have a consistent notation with the drag coefficient from Schiller-Naumann formulation, and made explicit the terms. Also updated notation to be consistent.

[...] The equation 3 defines the transport of particles. Here, C_d is the drag coefficient of the particle, which will depend on the implementation of the drag model used, in this work the Schiller-Naumann correlation was used since the particles are solid. The airflow velocity is represented by \mathbf{u} , and α_p is the volume fraction of the particles in the air, which in this case will be close to zero. The mass of the particle is denoted by m_p .

$$m_p \frac{d\mathbf{v}_p}{dt} = \frac{1}{2} C_d \rho A_p |\mathbf{v}_s| \mathbf{v}_s + m_p \mathbf{g} (1 - \alpha_p) + f_{\text{Brownian}} + f_{\text{lift}} + f_{\text{virtual mass}} \quad (3)$$

where C_d is the drag coefficient of the particle, ρ is the density of the continuous phase, $\mathbf{v}_s = \mathbf{u} - \mathbf{v}_p$ is the particle slip velocity, with \mathbf{u} being the instantaneous velocity of the continuous phase. The A_p is the projected area of the particle, \mathbf{g} is the gravitational acceleration, f_{Brownian} is the force due to Brownian motion, f_{lift} is the lift force, and $f_{\text{virtual mass}}$ is the virtual mass force.

Reference: Schiller, L., and Naumann, A. 1933 “Ueber die grundlegenden Berechnungen bei der Schw-erkraftaufbereitung”, VDI Zeits., 77(12), pp. 318–320.

Schiller, L. (1933). A drag coefficient correlation. Zeit. Ver. Deutsch. Ing., 77, 318-320.

2. The authors presented results for submicrometer-to-micrometer size of particles:

a. Authors need to adjust the length scale, including in the figures, to represent a particle of 0- μm .

The article presented used a log-normal distribution of particles with a mean of 0.42 μm . All the figures included the zero for reference, where the scale was linear and where it corresponded. In Figure 9 (originally Figure 8) of the manuscript, the 0- μm was also included.

b. The models were simulated as a two-way coupling for 100,000 particle density number and an average of 0.42- μm size particle at a 90 L/min flowrate to add robustness to the models, according to L327-328. However, the ratio of particle mass to fluid mass inside the HRT computational domain is not reported.

This is needed to aid establishing whether this is a dilute system (i.e., one-way coupling). And, if indeed, the system should be represented as a two-way:

The system should be represented as two-way due to the nature of the simulations and how the airflow is modeled. Since the simulation is transient, with the airflow modeled by a sin function, the ratio of volume displaced by the particles (V_p) related to the volume of fluid (V_f) is more significant than 10^{-6} . In Figure 5, we provide the volume occupied by each phase and report the volume fraction of particles (ϕ) in the caption. See answer to b.i. Below is the justification for two-way coupling having those values.

Figure 5: Volume occupied by particles and ROI volume (red box). The picture on the left is the simulation state at $t = 1s$ (end of inhalation), where the Volume fraction of Particles (ϕ) is $\phi = 5.22 \times 10^{-6}$. For the middle picture, the simulation state is at the beginning of the breathing cycle ($t = 0.1s$), and that particular ROI analyzed, the value of $\phi = 7.57 \times 10^{-6}$. Lastly, the right picture displays the simulation state in the nasal cavity at $t = 0.06s$, and the value of $\phi = 2.97 \times 10^{-5}$.

i. What are the contributions of particles on the dynamics of the fluid? And,

This question is addressed in great detail in the work by Elgobashi S. (1994), "On predicting particle-laden turbulent flows" we refer the Reviewer to that work for a more in-depth answer. However, as a summary, if the Particle Volume Fraction, defined as the ratio between the volume occupied by the particles and the volume of fluid + particles in that region, is between 10^{-6} and 10^{-3} , then two-way coupling must be used, since "the momentum transfer from the particles is large enough to alter the turbulence structure".

Reference: Elghobashi, S. On predicting particle-laden turbulent flows. Appl. Sci. Res. 52, 309–329 (1994). <https://doi.org/10.1007/BF00936835>

Moreover, two simulations for the same HRT geometry were analyzed, one without particles and the other with particles, using two-way coupling. By analyzing the difference in the pressure between the two simulations, it can be seen how the particles affected the global structure of the fluid.

ii. How, by quantification, is particle deposition impacted by the choice of coupling approach?

The decision on the particle coupling will affect deposition, and this behavior was also quantified in the past by Feng, Y., & Kleinstreuer, C. (2014). In their work "Micron-particle transport, interactions and deposition in triple lung-airway bifurcations using a novel modeling approach. In Journal of Aerosol Science (Vol. 71, pp. 1–15). Elsevier BV. <https://doi.org/10.1016/j.jaerosci.2014.01.003>". The authors decided to include a reference to this work on L300 of the original manuscript as follows:

More details on these results can be found in the Supplementary information, in Appendix I. This was also highlighted by Feng, Y., & Kleinstreuer, C. (2014). [reference], where differences in the particle deposition efficiency was quantified in simplified models of the HRT between one-way and two-way coupling.

iii. How the results of this paper are compared to the findings of "previous investigations", which should also be cited in L327?

This paper's results differ from those of other researchers who have investigated and cited throughout the manuscript, not only in the particle-flow coupling mechanisms but also in the inlet conditions used. To our knowledge, this is the only work that uses a heavy-breathing airflow of 90 L/min in the realistic lower respiratory tracts and a realistic total respiratory tract. Therefore, it would not be appropriate to compare the results of this paper to those of others since the conditions used in other works are different. Therefore, tracking the source for the difference between the results will be impossible due to the different multiple parameters involved in the simulations.

We modified L327 as follows:

Previous investigations [added references] have often limited their scope to one-way coupling, overlooking the complexity revealed by our more rigorous approach, and light-exercise respiratory breathing conditions.

3. Lastly, despite several forces were defined to influence the particle dynamics in this work, the specific contribution of the mechanisms of deposition or transport on the total deposition of the particles is not discussed. For instance, what are the contribution of Brownian forces on particle fate versus that of inertial forces or eddies?

First, we would like to thank Reviewer #2 for this insightful comment. We strongly agree that an extensive analysis of the specific contributions of various forces on particle deposition mechanisms can provide valuable insights. However, it is out of the scope of this work. Such analysis will require an extensive and rigorous investigation, which could be a separate journal article.

In the presented study, several forces that theoretically influence particle dynamics were considered, including Brownian motion, inertial forces, gravitational forces, lift forces, drag forces, and the effects of virtual mass. The theoretical basis for including Brownian motion is its significant impact on nano-sized particles, as is the case of the radioactive aerosols studied in this work, that have particle diameter distributions in the nano-diameter range.

According to the literature, Brownian forces dominate the motion of such particles in regions of low velocity (i.e., close to the walls), leading to enhanced diffusion and potentially more significant deposition. In contrast, inertial forces will be more influential for larger particles (greater than $1 \mu\text{m}$), as they influence how a particle will behave in the presence of flow obstructions or changes in flow direction.

Additionally, we have accounted for lift forces, caused by velocity gradients in the flow, and drag forces, which oppose particles' motion through a fluid. Moreover, virtual mass effects were also included to account for the added mass a particle must displace as it accelerates through a fluid, particularly during rapid changes in flow velocity (i.e., eddies).

A detailed parametric study analyzing the contributions of each force under varying conditions would be necessary to gain a more comprehensive understanding of how these forces affect particle deposition. This study would involve extensive computational simulations and further validation to isolate and quantify the effects of each force on particle deposition.

We appreciate the reviewer's suggestion and acknowledge its importance for future research. In the present study, we demonstrate the overall deposition trends and model particle motion as realistically as possible.

3 Reviewer #3

There are a lot of positives to this manuscript. It's nicely written. After careful review, I have the following concerns about the manuscript:

We sincerely thank Reviewer #3 for their review of our manuscript and for recognizing the positive aspects of our work. Reviewer #3 constructive feedback is highly appreciated as it guided our revisions to improve the manuscript.

We understand that there are concerns regarding our study's novelty, representativeness, and practical impact, as well as the need for thorough validation and clear recommendations for future research. In response to Reviewer #3 detailed comments, we have made the necessary revisions to address these issues and enhance the clarity and impact of our work.

Below, we provide a point-by-point response to each of Reviewer #3 comments, detailing the specific changes made to the manuscript:

1. The study's novelty may be questioned due to the existence of numerous prior works in related areas. I am not sure how this study will improve the knowledge of the field.

Although there are several different works that studied particle deposition in different regions of the HRT using CFD and CFPD, our work improves in the following areas:

- It introduces the first large-scale study on reconstructed HRT from CT scans.
- Is the first work on using two-way coupling for the particles and the fluid in various HRTs.
- It addresses multiple lower HRTs and also one full HRT. We, moreover, released all the geometries ready for CFPD, so other authors can benefit from already reconstructed and CFD/CFPD ready in the form of a Springer Nature figshare repository.

- A hybrid-automated workflow was presented and is available under reasonable request. That means that we are expediting the simulation setup and pre-processing steps to go from a CT scan to a CFD/CFPD simulation in less than an hour. Therefore, patient-specific studies or interventions will be possible with such a tool.
- We provide a coupling of the CFPD results to PHITS Monte-Carlo software to assess the damage of inhaled particle distributions. Although, this contribution still needs further refinement to be useful in field operations.
- Medium/Heavy-exercise breathing conditions were used, with a peak of 90 L/min in the respiratory cycle. Previous studies limited their work to resting/light-exercise breathing conditions.
- Modeling the respiratory cycle as a sin function to represent a more realistic inlet condition is something that only a few works have done in the past.

2. Reliance on generic mathematical models may not fully capture individual physiological variances.

The choice of such turbulence models was based on a trade-off between the computational complexity of the model and accuracy. Using LES would have been more accurate than relying on RANS turbulence models, but it is computationally too expensive to apply in many geometries. Therefore, based on previous literature, we opted for a RANS four-equation turbulence modeling that accounts for transitional regimes and can model both turbulent and laminar regimes, that is, $k - \omega$ SST LM or also sometimes referred to as $k - \omega$ SST $\gamma - Re_{\theta_t}$. Regarding the geometries of HRTs used, we presented automated tools to perform personalized CFD/CFPD assessment based on CT scans from individuals, aiming to capture individual physiological variances in the HRT.

3. Though large, the sample size does not represent global population diversity.

We agreed with Reviewer #3, but we looked for every possible open-source database that contain valid CT scans to reconstruct HRTs, using the EXACT'09 challenge and the TCIA database as resources. Future research should focus on expanding the database used to account for individuals who represent global population diversity well.

4. The practical impact on current clinical practices is not fully demonstrated.

The practical impact on current clinical practices comes from the semi-automatized simulation workflow, as delineated in Figure 10 (of the old manuscript, Figure 11 in the updated manuscript). The main innovation is based on speeding up the pre-processing process of going from a CT scan of an individual to a CFPD simulation. Usually, the reconstruction and pre-processing steps take several hours and human intervention. We developed scripts to automatize such a process, so in less than an hour (We achieved times less than 20 min. for experienced users), the CFPD simulation can be set up. The computational time of the CFPD simulation will depend on the resources available to perform the calculations. However, the positive is that the solvers used in this work are highly optimized and scale very well with the number of cores available for computation.

5. Computational models require thorough validation, with a more detailed discussion of potential errors.

In this regard, we added an extra validation step and more information on the previous validation done. Please refer to the answer to question #6 of Reviewer #1.

6. The study could be more explicit about the next steps and potential improvements, with clear recommendations for further research.

Thanks to this comment, we improved the conclusions on L338-L340 of the manuscript as follows:

~~Further venues are to explore in more depth the difference between idealized models from ICRP and our subject-specific modeling. More refinement to the presented algorithm is constantly done to automatize the process with minimum or no human intervention.~~

Ongoing efforts are focusing on improving the algorithms presented to fully automatize the process of going from a CT scan to a CFD or CFPD simulation, which will further reduce the time spent on pre-processing to a timescale in minutes. Moreover, future efforts focus on employing different initial source term particle distributions, accounting for electrostatic effects on the particle forces and the lower HRT used in the ICRP Publication 145 mesh phantoms. One limitation is that the adult-male phantom does not have a detailed HRT defined. Therefore, a more accurate mapping of the HRT is required for the ICRP Publication 145 phantom to account for realistic HRTs.

Point-by-point response to the referees' comment on manuscript COMMS-ENG-23-0605-A.

The point-by-point response to the referees' comments is organized in the following format:

- Black text: Original comments and questions from the reviewers.
- Blue text: Our responses to the referees.
- Red text: Original manuscript text.
- **Highlighted red text**: Added text to the resubmitted manuscript.
- ~~Strike-through red text~~: Deleted text from the original manuscript.

1 Reviewer #1

Thank the authors for their responses to my previous comments and revising their manuscript. All my questions and concerns have been properly addressed in the updated manuscript. The revised manuscript is significantly improved in quality, therefore the publication of this manuscript on Communications Engineering is recommended.

The authors wholeheartedly thank Reviewer #1 for their comments and finding the changes of significant improvement for the manuscript. The authors appreciate Reviewer #1's valuable time reviewing the updated manuscript.

2 Reviewer #2

Thank you for your comments and addressing some of my observations. Regarding (1.a): There are still some confusing narrative between CFD and CFPD. CFPD treats the particle as a discrete phase so it is purely Lagrangian based (in an Eulerian mesh). CFD when directed to particle studies treats this species as a gas phase, using the Eulerian method (i.e., the discrete method is not implemented), lines 61-64 showed some ambiguity in that regards. Lines 64 to 70 seemed unnecessary as it should be focused on the approach that better describes the dynamics between the fluid and the particles for the studied cases scenarios. Lastly, regarding (3), possible deposition mechanisms are not discussed in the latest manuscript. For instance, if particles can indeed deposit by Brownian diffusion, the particle drag model should account for that, and Stokes-Cunningham drag model becomes relevant. If some assumptions were made in the models, it should clearly be stated in the manuscript.

The authors are deeply grateful for the valuable time taken by Reviewer #2 to re-read the updated manuscript and provide an insightful review to improve the manuscript. Below, we provide a point-by-point response to the comments of Reviewer #2.

Regarding (1.a), the authors understand the source of the confusion and will include your comment for further clarification. However, it is worth pointing out that in the literature related to particle deposition using in-silico methods, CFPD is used for both Euler-Euler and Euler-Lagrange methods, being the first word the one used to explicit the coordinate system in which the fluid is solved (as you mentioned, in a purely Eulerian system) and the second word usually referring to the system in which the particles are solved. Certainly, if solved in an Eulerian description, there will not be any particle tracking information. When solved using an Eulerian framework, instead of particle deposition, researchers usually obtain a deposition enhancement factor (See, for example, Zhang, Z., & Kleinstreuer, C. (2004). Airflow structures and nanoparticle deposition in a human upper airway model. In *Journal of Computational Physics* (Vol. 198, Issue 1, pp. 178-210). Elsevier BV. <https://doi.org/10.1016/j.jcp.2003.11.034>). According to this comment, we modified the manuscript L61-L65 as follows:

In transitioning from CFD to CFPD, coupling the Navier-Stokes equations with the equations of motion for particles is critical for simulating fluid-particle interactions, accounting for the various mechanisms of particle deposition influenced by airflow and different particle sizes. When coupling the flow field with

particles, there are two primary approaches. The first approach is using a Lagrangian frame of reference, which treats particles as discrete points moving based on interactions within the flow field and other particles, by solving the equations of motion for each particle. The second approach is using an Eulerian frame of reference, which assumes particles are a continuous medium with representative properties, by solving a convection-diffusion equation. In both cases, the flow field is resolved on an Eulerian mesh description. In the following sections, the terms Euler-Lagrange and Euler-Euler refer to the method used to solve the motion of the particles, the first word in the term references how the flow field is solved, and the second word refers to the method used to solve the motion of particles.

On the other hand, there are different levels of complexity on how to model the particle and fluid interactions. The one-way coupled Euler-Lagrange [...]

Concerning (3), while noncontinuum slip effects on particles in the submicron level should be taken into account on the drag coefficient using the Cunningham correction factor, StarCCM+ does not provide such correction or any other correction for the drag coefficient, and only the Schiller-Naumann correlation can be used. Based on this, future work will focus on quantifying the differences in particle deposition by using the Cunningham slip factor correction and investigating the contribution of the Brownian motion to particle deposition. Thanks to this comment, L546-L547 of the manuscript was modified as follows.

Equation 5 defines the transport of particles, where C_d is the drag coefficient of the particle, which is dependent on the implementation of the drag model used; in this work, the Schiller-Naumman correlation was used since it is the only drag model available for solid particles in StarCCM+. It should be noted that the drag model used in this work does not consider the non-continuum slip effects on particles at the submicron level, which can be corrected by using the Cunningham slip correction factor.